# PUBLIC DATA-ASSISTED MIRROR DESCENT FOR PRIVATE MODEL TRAINING

## ABSTRACT

In this paper, we revisit the problem of using public data to improve the privacy/utility trade-offs for differentially private (DP) model training. Here, public data refers to auxiliary data sets that have no privacy concerns. We consider public training data sets that are from the *same distribution* as the private training data.

For convex losses, we show that a variant of Mirror Descent provides population risk guarantees which are independent of the dimension of the model ($p$). Specifically, we apply Mirror Descent with the loss generated by the public data as the *mirror map*, and using DP gradients of the loss generated by the private (sensitive) data. To obtain dimension independence, we require $G_Q^2 \leq p$ public data samples, where $G_Q$ is the Gaussian width of the smallest convex set $Q$ such that the public loss functions are 1-strongly convex with respect to $\|\cdot\|_Q$. We further show that our algorithm has a natural "noise stability" property: If in a bounded region around the current iterate, the public loss satisfies $\alpha_{\mathbf{v}}$-strong convexity in a direction $\mathbf{v}$, then using noisy gradients instead of the exact gradients shifts our next iterate in the direction $\mathbf{v}$ by an amount proportional to $1/\alpha_{\mathbf{v}}$ (in contrast with DP stochastic gradient descent (DP-SGD), where the shift is isotropic). Analogous results in prior works had to explicitly learn the geometry using the public data in the form of preconditioner matrices. Our method is also applicable to non-convex losses, as it does not rely on convexity assumptions to ensure DP guarantees.

We demonstrate the empirical efficacy of our algorithm by showing privacy/utility trade-offs on linear regression, deep learning benchmark datasets (WikiText-2, CIFAR-10, and EMNIST). We show that our algorithm not only significantly improves over traditional DP-SGD, which does not have access to public data, but also improves over DP-SGD on models that have been pretrained with the public data to begin with.

## 1 INTRODUCTION

Differentially Private Stochastic Gradient Descent (DP-SGD) (Song et al., 2013; Bassily et al., 2014; Abadi et al., 2016), and its variants (Kairouz et al., 2021b) have become the de facto standard algorithms for training machine learning models with differential privacy (DP) (Dwork et al., 2006). While DP-SGD is known to be optimal in terms of obtaining both optimal excess empirical risk (Bassily et al., 2014), and excess population risk (Bassily et al., 2020b) for convex losses, the obtained error guarantees suffer from an explicit polynomial dependence on the model dimension ($p$). This polynomial dependence significantly impacts the privacy/utility trade-off when $p \geq n_{\text{priv}}$, where $n_{\text{priv}}$ is the number of private training samples. Thus, even empirically, when DP-SGD is used to train large deep learning models, there is a significant drop in accuracy compared to the non-private counterpart (Papernot et al., 2020). In this paper, we revisit the problem of effectively using public data (i.e., data drawn from the same distribution as the private training data set, but without privacy concerns) to improve the privacy/utility trade-offs for DP model training. *Specifically, we design a central DP variant of mirror descent (Nemirovsky & Yudin, 1983) that uses the loss function generated by the public data as the mirror map and DP gradients on the private/sensitive data as the linear term, ensuring population risk guarantees for convex losses with no explicit dependence on dimensions as long as $n_{\text{pub}} \geq p$, where $n_{\text{pub}}$ is the number of records in the public data set.*

We show both theoretically and empirically that our DP variant of mirror descent, assisted with public data, can improve the privacy-utility trade-offs by effectively reducing the variance in the noise added to the gradients in DP model training. Our empirical results are on standard benchmark data sets like CIFAR-10, EMNIST, and WikiText-2.

**Learning Geometry with Mirror Maps:** Common to most DP model training algorithms, including DP-SGD, DP-FTRL (Kairouz et al., 2021b), and our algorithm, is a DP estimator of the gradient of the loss $\nabla_\theta \mathcal{L}(\theta_t; D_{\text{priv}}) = \sum_{d \in D_{\text{priv}}} \nabla_\theta \ell(\theta_t; d)$ generated by the private data set $D_{\text{priv}}$ at a given model state $\theta_t \in \mathbb{R}^p$. This DP estimator essentially adds isotropic Gaussian noise $\mathcal{N}(0, \sigma^2 \mathbb{I}_p)$ to $\nabla_\theta \mathcal{L}(\theta_t; D_{\text{priv}})$, where $\sigma$ depends on the privacy parameters $(\varepsilon, \delta)$ and the maximum allowable value of $\|\nabla_\theta \ell(\theta_t; d)\|_2$ (a.k.a. the clipping norm (Abadi et al., 2016)).[1] It is well known that for most learning tasks, the set of gradients vectors in $\mathcal{L}(\theta_t; D_{\text{priv}})$ are seldom isotropic (Gur-Ari et al., 2018; Agarwal et al., 2019). Hence, it is natural to wonder if the Gaussian noise in the DP estimator can be made to respect the geometry of the gradients. Prior works (Zhou et al., 2020; Asi et al., 2021; Kairouz et al., 2021a) have used public data ($D_{\text{pub}}$) to *explicitly* learn this geometry, mostly in the form of preconditioner matrices (Duchi et al., 2011) to be multiplied to the estimated noisy gradients. In this paper, we take an *implicit* approach towards respecting this geometry, by using the loss $\mathcal{L}(\theta; D_{\text{pub}})$ generated by the public data as the mirror map in classical mirror descent. As a first order approximation (formalized in Section 4), one can view it as doing DP-SGD on $\mathcal{L}(\theta; D_{\text{priv}})$ while using $\mathcal{L}(\theta; D_{\text{pub}})$ as a regularizer. This approach has the following advantages: (i) The information of the geometry is "free", i.e., one does not need to learn the preconditioner explicitly from the public data, (ii) Unlike prior works (Zhou et al., 2020; Kairouz et al., 2021a), one does not need to assume that the gradients of $\mathcal{L}(\theta; D_{\text{priv}})$ lie in a fixed rank subspace, (iii) The achieved excess population risk guarantees have better dependence on $n_{\text{pub}} = |D_{\text{pub}}|$ compared to prior results (Asi et al., 2021), and (iv) Because the geometry is implicitly defined, the implementation does not need to maintain an additional data structure for the preconditioner, and hence is much easier to implement. Empirically, under our best-effort comparison, our *baseline algorithm* improves over the state of the art (Asi et al., 2021).[2] We note that differentially private mirror descent has been considered before by Talwar et al. (2014) and Wang et al. (2017). Their results are not directly comparable to ours because (i) they do not have access to in-distribution public data (ii) as shown in Bassily et al. (2014), without public data, it is impossible to achieve the dimension independent bounds we achieve (iii) in our experiments we solve unconstrained optimization problems, but those works choose the mirror map based on the constraint set rather than the data set. We note that the utility bounds we prove in this paper also apply to a public data-assisted variant of the accelerated mirror descent algorithm considered in Wang et al. (2017).

**In-distribution vs. Out-of-distribution Public Data:** Prior works have considered both settings where the public data set $D_{\text{pub}}$ comes from the same distribution as the private data $D_{\text{priv}}$ (a.k.a. *in-distribution*) (Bassily et al., 2018a; Zhou et al., 2020; Kairouz et al., 2021a; Asi et al., 2021), and where the distributions are different (a.k.a. *out-of-distribution*) (Abadi et al., 2016; Papernot et al., 2016; 2018; Liu et al., 2021). In principle, *our algorithm can be used in out-of-distribution settings*, but our results in this paper are for the in-distribution case. In the in-distribution setting, it is typical that there are fewer public data samples available than private data samples – i.e., $n_{\text{pub}} \ll n_{\text{priv}}$ – as it is harder to obtain public data sets than ones with privacy constraints attached. In-distribution public data could come from either altruistic *opt-in* users (Merriman, 2014; Avent et al., 2017) or from users who are incentivized to provide such data (e.g., mechanical turks). Out-of-distribution public data may be easier to obtain but can have various degrees of freedom; e.g., the domains of private and public data may not be identical, the representation of some classes may vary, the distributions can be mean shifted, etc. It is usually hard to quantify these degrees of freedom to the extent that we can provide precise guarantees. Hence, we leave this aspect for future exploration, and work with the idealized assumption that the public data comes from the same distribution as the private data, or, at least, that the differences between these two distributions are not material.

**Design of Algorithms without Relying on Convexity for Privacy:** While most of our theoretical utility guarantees are specific to convex functions, our algorithm can be used seamlessly in non-convex settings. The main reason is that its DP guarantee does not rely on convexity. Prior work

---

[1] For the ease of presentation, at this point we do not consider the noise due to stochastic mini-batching.

[2] The implementation of Asi et al. (2021) is not publicly available. Since the algorithms can be sensitive to hyperparameter choices, for a fair comparison we directly consider the results quoted in (Asi et al., 2021).

that provided similar dimension-independent excess population risk guarantees under the the same conditions as ours (i.e., $n_{\mathsf{pub}} \geq p$) (Feldman et al., 2018) heavily relied on convexity to ensure DP and hence, cannot be used in non-convex settings.

**Choice of Benchmark for Empirical Comparison:** Mirror descent (Nemirovsky & Yudin, 1983; Hazan, 2019) as a first step optimizes the mirror map function. In our setting, this corresponds to pre-training on the public loss function $\mathcal{L}(\theta; D_{\mathsf{pub}})$ before running the DP optimization procedure on $\mathcal{L}(\theta; D_{\mathsf{priv}})$. Since pre-training on public data is intuitive and easy, we always compare to DP-SGD (and its variants) that have been pre-trained to convergence with the public loss. We show that our algorithm *outperforms* even pre-trained DP-SGD. To the best of our knowledge, ours is the only empirical work that compares to this strong (but fair) benchmark.

**Other Uses of Public Data in DP Learning:** The use of in-distribution public data has been extensively explored both theoretically and empirically. On the theoretical side, it has been shown (Alon et al., 2019; Bassily et al., 2020a) that a combination of private and public data samples can yield asymptotically better worst-case PAC learning guarantees than either on their own. Another line of work (Papernot et al., 2016; 2018; Bassily et al., 2018b; Dwork & Feldman, 2018; Nandi & Bassily, 2020) considers public data that is unlabelled, but otherwise comes from the same distribution as the private data; the primary goal is to use the private data to generate labels for the public data, which can then be used arbitrarily. So far only two papers have considered out-of-distribution data. Bassily et al. (2020c) assume that whether a data record is public or private depends on its label; e.g., the public data may contain many negative examples, but few positive examples. They show that halfspaces can be learned in this model. Liu et al. (2021) consider synthetic data generation and provide guarantees that depend on the Rényi divergences between the public and private distributions. Abadi et al. (2016) and Tramer & Boneh (2020) provided techniques to effectively use out-of-distribution public data for pre-training for DP-SGD. However, they did not consider techniques to improve a pre-trained model using private and public data, which is the focus of our work.

### 1.1 PROBLEM FORMULATION

Consider the classic differentially private stochastic convex optimization (DP-SCO) (Chaudhuri et al., 2011; Bassily et al., 2014; 2019; 2020b) setting. Let $\tau$ be a distribution over a fixed domain $\mathcal{D}$. Given a data set $D \in \mathcal{D}^*$ drawn i.i.d. from $\tau$, and a loss function $\ell_{\mathsf{priv}} : \mathbb{R}^p \times \mathcal{D} \to \mathbb{R}$, the objective is to approximately solve $\arg\min_{\theta \in \mathcal{C}} \mathbb{E}_{d \sim \tau} [\ell_{\mathsf{priv}}(\theta; d)]$, while preserving DP. Here, $\mathcal{C} \subseteq \mathbb{R}^p$ is the constraint set. Usually one solves the SCO problem via empirical risk minimization (ERM), i.e., $\theta^{\mathsf{priv}} \in \arg\min_{\theta \in \mathcal{C}} \mathcal{L}(\theta; D)$, where $\mathcal{L}(\theta; D) = \frac{1}{|D|} \sum_{d \in D} \ell_{\mathsf{priv}}(\theta; d)$, and then uses $\theta^{\mathsf{priv}}$ as a proxy. Up to a dependence on dimensionality $p$, in the DP setting, a direct translation from ERM to the SCO setting provides the optimal rates (Bassily et al., 2014; 2019; 2020b). Since we will strive for dimension-independent convergence rates, we thus safely ignore the issue of non-optimality due to dimensionality, and treat the ERM and SCO problems as almost equivalent.

In this paper, we consider the DP-SCO setting with *heterogeneous data*, where there are two data sets $D_{\mathsf{priv}}$ (with $n_{\mathsf{priv}}$ samples) and $D_{\mathsf{pub}}$ (with $n_{\mathsf{pub}}$ samples) drawn i.i.d. from the *same distribution*. The private data set $D_{\mathsf{priv}}$ consists of records from individuals who require privacy protection, whereas the public data set $D_{\mathsf{pub}}$ does not require any privacy protection. Since obtaining such data can be expensive, for our empirical evaluation, we assume $n_{\mathsf{pub}}$ is a small fraction of $n_{\mathsf{priv}}$ (e.g., $< 5\%$).

We will also use a separate public loss function $\ell_{\mathsf{pub}}$. For our theoretical analysis, we will assume $\ell_{\mathsf{priv}}$ is convex and $L$-Lipschitz, whereas $\ell_{\mathsf{pub}}$ is strongly convex (both with respect to the $\ell_2$-norm). In practice, one will likely choose $\ell_{\mathsf{priv}} = \ell_{\mathsf{pub}}$, but clip the gradients of $\ell_{\mathsf{priv}}$ for privacy. In general, $\ell_{\mathsf{pub}}$ can be arbitrary, but for the analysis, we will assume that $\ell_{\mathsf{pub}}$ and $\ell_{\mathsf{priv}}$ roughly share a minimizer and that $\nabla \ell_{\mathsf{pub}}(\theta; d)$ has bounded variance (see Assumption 3.1 for a formal statement). We refer the reader to Appendix A for a reference for the notation used throughout the paper.

### 1.2 OUR CONTRIBUTIONS

**Dimension Independent Excess Population Risk for Constrained ERMs:** The standard private SCO excess empirical loss bound for a constraint set $\mathcal{C}$ is: $O\left( L \left\| \mathcal{C} \right\|_2 \cdot \frac{\sqrt{p \log(1/\delta)}}{\varepsilon n_{\mathsf{priv}}} \right)$, where $L$ is the

Lipschitz constant of the loss function and $n$ is the number of samples in the dataset. If we pre-train on public data and are able to find a solution within distance $O\left(\frac{1}{\sqrt{p}}\right)$ of the optimal solution, we can set $\mathcal{C}$ to be the ball of radius $O\left(\frac{1}{\sqrt{p}}\right)$ centered on the pre-trained model and obtain a similar dimension-independent error rate. There are a couple of issues with this approach: (i) It bakes in the utility assumption about the constraint set into the algorithm. This is unsatisfactory because if we are unlucky (due to stochasticity or choice of hyperparameters) and the optimal solution does not lie within this ball, using this choice of $\mathcal{C}$ may restrict us to a range of poor solutions that we cannot escape even if the number of private samples is very large, and (ii) It forces one to operate with projected gradient descent which may significantly impact accuracy in settings where the utility assumptions may not hold, e.g., in deep learning.

We improve upon this by showing that given $n_{\mathsf{pub}}$ public samples in addition to $n_{\mathsf{priv}}$ private samples, using our PDA-DPMD (*Public Data-Assisted Differentially Private Mirror Descent*) algorithm, a variant of mirror descent that implicitly reshapes the noisy gradients according to the public loss function, we can achieve excess empirical loss $O\left(LV \cdot \frac{G_Q\sqrt{\log(1/\delta)}}{\varepsilon n_{\mathsf{priv}}\sqrt{n_{\mathsf{pub}}}}\right)$. The bound is proven in Theorem 3.2. Here, $V$ is a measure of the variance of the gradients of $\ell_{\mathsf{pub}}$, and $G_Q$ is the Gaussian width of the smallest set $Q$ such that the public loss functions are 1-strongly convex with respect to $\|\cdot\|_Q$. In particular, $G_Q$ is at most $\sqrt{p}$ if the public loss functions are 1-strongly convex, but can be constant if e.g. the public loss functions have a much larger strong convexity parameter in all but a constant number of basis directions. Note that if we have $n_{\mathsf{pub}} = G_Q^2$ public samples, then the $1/\sqrt{n_{\mathsf{pub}}}$ and $G_Q$ terms cancel out, i.e. this error bound has no explicit dependence on dimension. Using standard techniques, we can turn this into a dimension-independent excess population loss bound (see Theorem 3.5), again assuming $n_{\mathsf{pub}} \geq G_Q^2$. In addition, to the best of our knowledge, ours is the first work on augmenting private training with public data to show a theoretical improvement over DP-SGD (here the dependence $\frac{1}{\sqrt{n_{\mathsf{pub}}}}$) due to pre-training on public data. In particular, we show pre-training improves the standard DP-SGD bounds even under a totally isotropic geometry.

**Local Noise-stability of Mirror Descent:** We show that in addition to achieving better excess population loss bounds, PDA-DPMD has the following "local noise-stability" property: If in a bounded region around the current iterate, the public loss satisfies $\alpha_{\mathbf{v}}$-strong convexity in a direction $\mathbf{v}$, then using noisy gradients instead of the exact gradients shifts our next iterate in the direction $\mathbf{v}$ by an amount proportional to $1/\alpha_{\mathbf{v}}$ (see Theorem 3.6). That is, PDA-DPMD effectively rescales the effect of adding noise in any direction to be inversely proportional to the curvature in that direction. Note that this is in spite of the fact that for privacy, the noise we add to the gradients is usually isotropic. Furthermore, this rescaling is done purely as a function of the public loss function used by PDA-DPMD. In other words, a practitioner implementing PDA-DPMD simply needs to choose an appropriate loss function, and PDA-DPMD will "automatically" rescale the effects of noise to match the loss function's curvature.

**Empirical Evaluation:** On both synthetic and real-world benchmark data sets, we show that PDA-DPMD outperforms DP-SGD, even when they are pre-trained on the public data set. We provide two sets of experiments. First, a linear regression on a synthetic data set which closely matches the utility assumptions in the theoretical analysis. Second, we provide results on standard deep learning benchmark data sets (WikiText-2, CIFAR-10, and EMNIST).

In Section 3, we consider using DP-SGD and PDA-DPMD to solve a least squares linear regression problem on a synthetic data set generated via the process $b_i \sim N(\langle \mathbf{a}_i, \theta^* \rangle, \sigma^2)$, where $\theta^* \in \mathbb{R}^p$ is the true model. The feature vectors $\mathbf{a}_i$'s are drawn i.i.d. from some fixed distribution. We fix the number of private data samples, and set the number of public samples to be a fixed constant times the dimension ($p$). We observe that as expected, public data allows us to substantially improve the error in two ways: (i) **Pre-training**: DP-SGD initialized from a model pre-trained on public data has nearly-constant mean-squared error, whereas DP-SGD from a random initialization has mean-squared error scaling with the dimension, and (ii) **Adapting to geometry using public loss**: While DP-SGD initialized from a pre-trained model already achieves near-constant loss, we also observe that PDA-DPMD outperforms DP-SGD due to its error's dependence on the Gaussian width $G_Q$ rather than the dimension. We note that the observed improvement is "automatic" once we choose the mean-squared error to be the loss function.

We also conduct experiments in Section 4 on two real world tasks: next word prediction on WikiText-2, and image classification on CIFAR-10 and EMNIST (ByMerge split). We consider 4% of the original training data as public and pretrain on it. On all datasets, we can observe that an approximate version of PDA-DPMD outperforms DP-SGD in terms of test loss. On CIFAR-10, the improvement is more than 5%; on EMNIST, 7%; on WikiText-2, log perplexity is improved by more than 0.3%, which is a notable improvement for perplexity.

For the deep learning experiments, running PDA-DPMD was computationally expensive. We derive a first-order approximation that can be viewed as regular DP-SGD on a convex combination of private and the public losses, i.e., $\alpha \nabla \mathcal{L}(\theta; D_{\mathsf{priv}}) + (1 - \alpha) \nabla \mathcal{L}(\theta; D_{\mathsf{pub}}), \alpha \in [0, 1]$, where $\nabla \mathcal{L}(\theta; D_{\mathsf{priv}})$ is privatized by clipping and adding noise. This approximation makes the running time of our algorithm comparable to DP-SGD when run on the data set $D_{\mathsf{priv}} \cup D_{\mathsf{pub}}$.

## 2 BACKGROUND

**Differential Privacy:** Differential Privacy (DP) – originally defined by Dwork et al. (2006) – is a formal method for quantifying the privacy leakage from the output of a data analysis procedure. A randomized algorithm $M : \mathcal{D}^* \to \mathcal{Y}$ is $(\varepsilon, \delta)$-DP if, for all neighbouring dataset pairs $D, D' \in \mathcal{D}^*$ and all measurable sets of outputs $S \subseteq \mathcal{Y}$, we have

$$\mathbb{P}\left[M(D) \in S\right] \leq e^\varepsilon \cdot \mathbb{P}\left[M(D') \in S\right] + \delta.$$

We define two datasets to be neighbouring if they differ only by the *addition or removal of one person's record*. We ensure differential privacy by adding Gaussian noise to functions of bounded sensitivity. In particular, if $\ell$ is $L$-Lipschitz in its first parameter, then $\|\nabla_\theta \ell(\theta; d)\|_2 \leq L$ for all $\theta$ and $d \in \mathcal{D}$. Thus adding noise drawn from $\mathcal{N}(0, \sigma^2 \cdot \mathbb{I}_p)$ to the sum $\sum_i \nabla_\theta \ell(\theta, d_i)$ over people's records satisfies DP, where $\sigma$ scales with $L$ and the desired privacy parameters. The composition and postprocessing properties of differential privacy ensure that, as long as each step in our iterative algorithm satisfies differential privacy, then so does the overall system. We refer the reader to (Dwork & Roth, 2014) for further details of the standard privacy analysis of algorithms like ours.

Due to space constraints, we defer background on Mirror Maps and Gaussian Width to Appendix B.

## 3 PUBLIC DATA ASSISTED DIFFERENTIALLY PRIVATE MIRROR DECENT

In this section, we present our algorithm, PDA-DPMD, dimension-independent excess empirical and population loss bounds for PDA-DPMD (Theorems 3.2 and 3.5), and a "noise stability" property (Theorem 3.6). We defer all the proofs in this section to Appendix C.

**Algorithmic Description:** Our main algorithm, Public Data-Assisted Differentially Private Mirror Descent (PDA-DPMD), is given as Algorithm 1. PDA-DPMD is a variant of mirror descent using noisy gradients, but we also pre-train on public data and use the public loss as our mirror map $\Psi$.

---

**Algorithm 1** Public Data-Assisted Differentially Private Mirror Descent (PDA-DPMD)

---

**Input:** Public/private datasets $D_{\mathsf{pub}}, D_{\mathsf{priv}}$ of sizes $n_{\mathsf{pub}}, n_{\mathsf{priv}}$, private/public loss functions $\ell_{\mathsf{priv}}$, $\ell_{\mathsf{pub}}$, privacy parameters $(\varepsilon, \delta)$, number of iterations $T$, learning rate $\eta : \{0, 1, \dots, T - 1\} \to \mathbb{R}^+$, constraint set: $\mathcal{C}$, clipping norm $L$: an upper bound on $\max_{\theta \in \mathcal{C}} \|\nabla \ell_{\mathsf{priv}}(\theta)\|_2$

1: $\Psi(\theta) := \frac{1}{n_{\mathsf{pub}}} \sum_{d \in D_{\mathsf{pub}}} \ell_{\mathsf{pub}}(\theta; d), \theta_0 \leftarrow \arg\min_{\theta \in \mathcal{C}} \Psi(\theta), \sigma^2 \leftarrow \frac{8L^2 T \log(1/\delta)}{(\varepsilon n_{\mathsf{priv}})^2}$
2: **for** $t = 0, \dots, T - 1$ **do**
3: $\quad \mathbf{g}_t \leftarrow \frac{1}{n_{\mathsf{priv}}} \sum_{d \in D_{\mathsf{priv}}} \nabla \ell_{\mathsf{priv}}(\theta; d)$
4: $\quad \theta_{t+1} \leftarrow \arg\min_{\theta \in \mathcal{C}} \left[\eta_t \langle \mathbf{g}_t + \mathbf{b}_t, \theta \rangle + B_\Psi(\theta, \theta_t)\right]$, where $\mathbf{b}_t \sim \mathcal{N}(0, \sigma^2 \cdot \mathbb{I}_p)$
5: **end for**
6: **return** $\theta_{\mathsf{priv}} := \frac{1}{T} \sum_{t=1}^{T} \theta_t$

---

Note that Line 4 of PDA-DPMD is equivalent to the following: Choose $\theta_{t+1/2}$ to be the point such that $\nabla \Psi(\theta_{t+1/2}) = \nabla \Psi(\theta_t) - \eta(\mathbf{g}_t + \mathbf{b}_t)$, and then use the Bregman projection $\theta_{t+1} =$

$\arg\min_{\theta \in \mathcal{C}} B_\Psi(\theta, \theta_{t+1/2})$. Intuitively, PDA-DPMD is similar to DP-SGD, with the main difference being we apply the gradient steps to $\nabla\Psi(\theta)$ rather than to $\theta$ itself. Note that PDA-DPMD reshapes the gradient and noise *automatically* given $\ell_{\mathsf{pub}}$ and $D_{\mathsf{pub}}$. In contrast, e.g., private Ada-Grad implementations (Kairouz et al., 2020; Asi et al., 2021) assume knowledge of the geometry of the loss function has already been learned prior to running their algorithms. Also, for an appropriate choice of $\Psi$, one can recover an algorithm that projects the private gradients to a low-dimensional subspace as in the algorithms of Zhou et al. (2020) and Kairouz et al. (2020).

**Dimension Independent Empirical Risk Bounds for Convex Losses:** We bound the excess empirical loss on the public loss function $\ell_{\mathsf{pub}}$, compared to the private *population* minimizer $\theta^*$ (rather than the empirical minimizer). This is because the empirical minimizer $\theta_{\mathsf{emp}}$ of the private loss function could be far away from $\theta^*$, and in turn $\nabla\Psi(\theta_{\mathsf{emp}})$ could be much larger in expectation than $\nabla\Psi(\theta^*)$. Our PDA-DPMD excess empirical loss bound will be in terms of $\nabla\Psi(\theta)$, where $\theta$ is the point we are comparing to, so it is preferable to use $\theta = \theta^*$ for this reason.

We will use the following "bounded variance" assumption on the distribution of the datasets and the public loss function:

**Assumption 3.1.** *For some minimizer $\theta^* \in \arg\min_{\theta \in \mathcal{C}} \mathbb{E}_{d \sim \tau}[\ell_{\mathsf{priv}}(\theta; d)]$ we have that $\theta^*$ is also the minimizer of $\mathbb{E}_{d \sim \tau}[\ell_{\mathsf{pub}}(\theta; d)]$ in $\mathcal{C}$ and $\mathbb{E}_{d \sim \tau}\left[\|\nabla\ell_{\mathsf{pub}}(\theta^*; d) - \mathbb{E}_{d \sim \tau}[\nabla\ell_{\mathsf{pub}}(\theta^*; d)]\|_2^2\right] \leq V^2$. In particular, this implies*

$$\mathbb{E}_{D \sim \tau^{n_{\mathsf{pub}}}}\left[\left\|\frac{1}{n_{\mathsf{pub}}}\sum_{d \in D}\nabla\ell_{\mathsf{pub}}(\theta^*; d) - \mathbb{E}_{d \sim \tau}[\nabla\ell_{\mathsf{pub}}(\theta^*; d)]\right\|_2^2\right] = O\left(\frac{V^2}{n_{\mathsf{pub}}}\right).$$

We note that while Assumption 3.1 is written as generally as possible and thus captures scenarios where $\ell_{\mathsf{pub}}$ and $\ell_{\mathsf{priv}}$ could potentially be very different loss functions, the reader can think of them as differing only slightly. Indeed, Assumption 3.1 captures several scenarios we might see in practice, such as (i) $\ell_{\mathsf{pub}} = \ell_{\mathsf{priv}}$ (which can occur if $\|\mathcal{C}\|_2$ is small), (ii) $\ell_{\mathsf{priv}}$ is the clipped version of $\ell_{\mathsf{pub}}$ (see e.g., Song et al. (2021) for a discussion on the effects of clipping on the loss function), and (iii) $\ell_{\mathsf{pub}}$ is $\ell_{\mathsf{priv}}$ but with a regularizer added. Our empirical loss bound now follows by using Assumption 3.1 to control the Bregman divergence between the initial iterate and the population minimizer:

**Theorem 3.2.** *Suppose the private loss function $\mathcal{L} := \frac{1}{n_{\mathsf{priv}}}\sum_{d \in D_{\mathsf{priv}}}\ell_{\mathsf{priv}}(\theta; d)$ is $L$-Lipschitz and convex. Suppose $\ell_{\mathsf{pub}}$ is $\alpha$-strongly convex, and let $Q$ be the minimal convex body containing the origin such that each $\ell_{\mathsf{pub}}(\theta; d)$ is 1-strongly convex with respect to the Minkowski norm $\|\cdot\|_Q$ (defined as $\|x\|_Q = \min\{c \in \mathbb{R}_{\geq 0}|x \in cQ\}$). Then PDA-DPMD is $(\varepsilon, \delta)$-differentially private with respect to the private database $D_{\mathsf{priv}}$ and choosing $\eta_t = \eta$ for all $t$ we have:*

$$\mathbb{E}_{D_{\mathsf{pub}} \sim \tau^{n_{\mathsf{pub}}}}[\mathcal{L}(\theta_{\mathsf{priv}})] - \mathcal{L}(\theta^*) \leq \frac{V^2}{2\alpha\eta T n_{\mathsf{pub}}} + \eta \cdot O(L^2\|Q\|_2^2 + \sigma^2(G_Q^2 + \|Q\|_2^2)).$$

The above bound is scale-invariant, so to simplify the presentation of this section, we assume, without loss of generality, that $\alpha = 1$ (this also implies $Q$ is contained within the unit $\ell_2$-ball, i.e. $\|Q\|_2 \leq 1$). By rescaling $\Psi$ and $\eta$ appropriately, we do not affect the behavior of PDA-DPMD, but get that $\Psi$ is 1-strongly convex.

By chaining the following lemma with Theorem 3.2, we get an excess empirical loss bound with respect to the sample minimizer rather than the population minimizer as desired.

**Lemma 3.3.** *Let $\tau$ be a distribution over $\mathcal{D}$, $\ell : \mathcal{C} \times \mathcal{D} \to \mathbb{R}$ be a function such that $\ell(\theta; d)$ is $L$-Lipschitz and convex in $\theta$ for any fixed $d \in supp(\tau)$. Let $\theta^*$ be the minimizer of $\mathbb{E}_{d \sim \tau}[\ell(\theta; d)]$. Then, we have $\mathbb{E}_{D \sim \tau^n}[\mathcal{L}(\theta^*; D) - \min_{\theta \in \mathcal{C}}\mathcal{L}(\theta; D)] \leq \frac{L\|\mathcal{C}\|_2}{\sqrt{n}}$.*

The lemma follows by using convexity and a bound on the expected $\ell_2$-norm difference between the gradient of the empirical and population losses.

**Excess Population Risk of PDA-DPMD:** We now translate our excess empirical loss bound to a excess population loss. We use Lemma F.5 of Bassily et al. (2014), restated in Lemma 3.4 for convenience, which provides a black-box translation from empirical loss to population loss:

**Lemma 3.4.** *For any $(\varepsilon, \delta)$-DP algorithm for minimizing $\frac{1}{n_{\mathsf{priv}}}\sum_{d \in D_{\mathsf{priv}}}\ell(\theta; d)$ over $\mathcal{C}$, the expected excess population loss exceeds the expected excess empirical loss by $O(L\|\mathcal{C}\|_2\varepsilon + \|\mathcal{C}\|_2^2\delta)$.*

Given this lemma, it is straightforward to derive excess population loss bounds:

**Theorem 3.5.** *For* $\eta = \frac{V}{L\sqrt{Tn_{\text{pub}}}}, T = \frac{\varepsilon^2 n_{\text{priv}}^2}{G_Q^2 \log(1/\delta)}$, *and setting* $\varepsilon = \frac{\sqrt{VG_Q}\log^{1/4}(1/\delta)}{\sqrt{n_{\text{priv}}}n_{\text{pub}}^{1/4}\|\mathcal{C}\|_2}$, *the expected population loss of PDA-DPMD is*

$$O\left(\frac{L\sqrt{V\|\mathcal{C}\|_2}\sqrt{G_Q}\log^{1/4}(1/\delta)}{\sqrt{n_{\text{priv}}}n_{\text{pub}}^{1/4}} + \frac{L\|\mathcal{C}\|_2}{\sqrt{n_{\text{priv}}}} + \|\mathcal{C}\|_2^2\,\delta\right).$$

Theorem 3.5 follows immediately from Theorem 3.2, Lemma 3.3, and Lemma 3.4. Note that if $n_{\text{pub}} \geq G_Q^2$, which is at most $p$, then the above bound has no explicit dependence on dimension. For comparison, if we were to only train on public data, the standard non-private excess population loss bound has dependence $O(1/\sqrt{n_{\text{pub}}})$ (and no dependence on dimension). So in the regime where $n_{\text{pub}} \approx G_Q^2$ and $n_{\text{priv}} \gg n_{\text{pub}}$, our bound is asymptotically much better than the baseline of training only on public examples. In Appendix D we show what values $Q$ and $G_Q$ take on for some standard stochastic convex optimization problems to help the reader understand the bound in Theorem 3.5.

**Local Stability Properties of PDA-DPMD:** If the public loss function has a Hessian everywhere that has the same eigenvectors regardless of location (but perhaps different eigenvalues), we can bound the impact of adding noise to the gradient on each update in mirror descent:

**Theorem 3.6.** *Suppose for the public loss function* $\Psi$, *its Hessian is defined everywhere, and for a fixed orthonormal basis* $\{\mathbf{v}_i\}_i$, *the Hessian at every point can be written as* $\sum_i w_i(\theta)\mathbf{v}_i\mathbf{v}_i^\top$ *for scalar functions* $w_i : \mathbb{R}^p \to \mathbb{R}^+$ *such that for all* $i, \theta$, *we have* $w_i(\theta) \geq \alpha$. *Fix an iteration* $t$ *as well as private gradient* $\mathbf{g}_t$ *in PDA-DPMD. Let* $\theta^*$ *be the value of* $\theta_{t+1}$ *after performing the mirror descent update with* $\mathbf{b}_t = \mathbf{0}$ *at iteration* $t$, *and let* $\{\widetilde{w}_i\}_i, c \geq 0$ *be such that for each* $i$ *the smallest value of* $w_i(\theta)$ *in the ellipsoid* $E := (\sum_i \frac{1}{\widetilde{w}_i}\mathbf{v}_i\mathbf{v}_i^\top)B_R$ *(where* $B_R$ *is the* $\ell_2$ *ball of radius* $R := \eta(1+c)\sqrt{p}\sigma)$, *centered at* $\theta^*$, *is at least* $\widetilde{w}_i$. *Then for any (unit) direction* $\mathbf{v} = \sum_i a_i\mathbf{v}_i$,

$$\mathbb{E}\left[|\langle\hat{\theta} - \theta^*, \mathbf{v}\rangle|\right] \leq \eta\sigma\left[\sqrt{\frac{2}{\pi}} \cdot \sqrt{\sum_i \left(\frac{a_i}{\widetilde{w}_i}\right)^2} + \frac{3(1+c)^2}{2\alpha} \cdot e^{-c^2 p/2}\right],$$

*where* $\hat{\theta}$ *is the value of the next iterate* $\theta_{t+1}$ *if noise is added.*

At a high level, Theorem 3.6 follows by observing that $\hat{\theta}-\theta^*$ can be expressed as the inverse Hessian of $\Psi$ evaluated at some point $\widetilde{\theta}$ times the noise vector $\mathbf{b}_t$. If the magnitude of $\mathbf{b}_t$ is small enough, $\widetilde{\theta}$ stays within $E$ and we can lower bound the inverse Hessian by $\frac{1}{\widetilde{w}_i}\mathbf{v}_i\mathbf{v}_i^\top$. In the low-probability event $\mathbf{b}_t$'s magnitude is large enough, we can instead lower bound the inverse Hessian by $\frac{1}{\alpha}\mathbb{I}_p$.

Note that the condition $w_i(\theta) \geq \alpha$ can be enforced by adding an $\ell_2$-regularizer to the public loss function (since mirror descent only cares about differences in the gradients of the public loss function, the private training phase of PDA-DPMD behaves the same regardless of where this regularizer is centered). In contrast for DP-SGD, $\mathbb{E}\left[|\langle\hat{\theta} - \theta^*, \mathbf{v}\rangle|\right] = \sqrt{2/\pi} \cdot \eta\sigma$ for any direction $\mathbf{v}$.

**Validation of Dimension-independence on Synthetic Linear Regression:** To corroborate our theoretical results with empirical validation, we consider the linear regression problem with mean squared error loss: $\frac{1}{n_{\text{priv}}}\|\mathbf{X}\theta - \mathbf{y}\|_2^2$. We vary the dimensionality of the problem $p$ from 500 to 6000. For each $p$, we generate 10,000 private samples and $1.5p$ public samples. The optimal $\theta^*$ is sampled from $\mathcal{N}(0, \mathbb{I}_p)$. To introduce a non-isotropic geometry, we sample the feature vector $\mathbf{x}_i$ such that 40 of the first $p/5$ features and 80 of the last $4p/5$ features, chosen uniformly at random, are set to $0.05$, and the rest of the features are set to 0. In this way, the expected $\ell_2$-norm of each feature vector (and in turn each gradient) is dimension-independent, and thus the effects of clipping should not vary with $p$. The predicted variable $y_i$ is sampled from $\mathcal{N}(\theta^* \cdot \mathbf{x}_i, 0.01)$ so that the population mean squared error loss is always 0.01, i.e. independent of dimension. Since the norms of the gradients and number of private samples are dimension-independent, our error bound is proportional to $G_Q/\sqrt{p}$, i.e. we do not expect it to vary much with dimension. We set $\varepsilon = 1, \delta = 10^{-5}$.

We consider three algorithms: (i) standard DP-SGD with a "cold start", i.e. using a random initialization, (ii) standard DP-SGD with a "warm start" on the model pre-trained with public data, and (iii) PDA-DPMD after pre-training on public data.

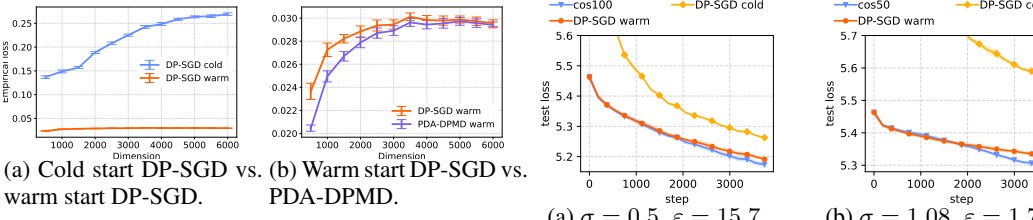

(a) Cold start DP-SGD vs. warm start DP-SGD.

(b) Warm start DP-SGD vs. PDA-DPMD.

(a) $\sigma = 0.5$. $\varepsilon = 15.7$.

(b) $\sigma = 1.08$. $\varepsilon = 1.71$.

Figure 1: Comparisons of the empirical loss on synthetic linear regression data. The mean and error bars for a 95% confidence interval over 20 runs are plotted. The optimal population loss is $0.01$.

Figure 2: WikiText-2. 4% public data. Validation and test loss. Averaged over 3 runs. Full plots available at Appendix F.3.

Figure 1a shows the empirical loss of cold- and warm-start DP-SGD. Since the dimension-dependent loss of DP-SGD is the excess empirical loss, we report empirical rather than population loss. As expected, pre-training with public data allows DP-SGD to achieve nearly dimension-independent error. Figure 1b compares warm-start DP-SGD and PDA-DPMD. The loss of PDA-DPMD is never worse than that of warm-start DP-SGD, and can be substantially lower for smaller dimensions. We observed that the ratio of the maximum and minimum eigenvalues of the Hessian $\mathbf{X}^\top \mathbf{X}$ decreases as $p$ increases, which means that the Hessian has poorly-concentrated eigenvalues at small $p$ but gets closer to the identity matrix as $p$ increases. Since PDA-DPMD recovers warm start DP-SGD when the Hessian is the identity, we can expect that PDA-DPMD obtains less of an advantage over DP-SGD as the Hessian gets closer to the identity. We provide additional details in Appendix E.

## 4 EMPIRICAL EVALUATION ON BENCHMARK DATA SETS

**First-order Approximation to Mirror Descent:** In practice, the Mirror Descent (MD) step in Line 4 of Algorithm 1 can be computationally expensive. For settings where (i) the problem is *unconstrained*, i.e., $\mathcal{C} = \mathbb{R}^p$ and (ii) the public loss function $\Psi(\theta)$ may not be strongly convex with respect to the $\ell_2$-norm, we can instead use the following more efficient approximation:

$$\theta_{t+1} \leftarrow \theta_t - \eta_t \left( \alpha_t(\mathbf{g}_t + \mathbf{b}_t) + (1 - \alpha_t)\nabla\Psi(\theta_t) \right), \tag{1}$$

where $\eta_t$ is the learning rate, and $\alpha_t \in [0, 1]$ balances the weight of private and public gradient. The derivation of this formula is in Appendix F.1. Notice that $\alpha_t = 1$ corresponds to DP-SGD on private data only. In our experiment, we decrease $\alpha_t$ with a cosine schedule, i.e. $\alpha_t = \cos(\pi t/(2K))$ where $K$ is a hyperparameter that controls how fast $\alpha_t$ decays. In practice, instead of computing $\mathbf{g}_t$ and $\nabla\Psi(\theta_t)$ using all the private and public data, we can estimate them with stochastic gradients.

Now, we demonstrate the efficacy of our technique (Algorithm 1) with the update step in (1) on two real world tasks across three benchmark data sets: next word prediction on WikiText-2 (Merity et al., 2017), and image classification on CIFAR-10 (Krizhevsky, 2009) and EMNIST (ByMerge split) (Cohen et al., 2017). For each dataset, we randomly assign 4% of the original training data as public, and the rest as private. We do not consider a larger amount of in-distribution public data as that could make the problem trivial. We first pre-train on the public data, then use Algorithm 1 with update rule (1). We compare our algorithm with two baselines: "cold-start" DP-SGD, which uses the private data only, and "warm-start" DP-SGD, which pre-trains the model with public data and then fine-tunes with the private data. We demonstrate an increased benefit from the public data over and above just pre-training, which to our knowledge, has not been achieved in prior work. For WikiText-2, we use an LSTM model from Asi et al. (2021). For CIFAR-10 and EMNIST, we use network architectures considered in prior works (Papernot et al., 2020; Kairouz et al., 2021b). See Appendix F.2 for more details.

**Empirical Evaluation on WikiText-2:** Our setup mainly follows Asi et al. (2021). As a preprocessing step, we take the top 7,999 most frequent words and convert the rest into a special token representing unknown word. The data set is then split into 48,764 length-35 sequences, and we consider sequence-level privacy here. After pre-training, we fine-tune the model with batch size 250 for 20 epochs. We search for optimal $K$ in $\{100, 200, 500\}$. For two different privacy levels, $\varepsilon = 15.7$ and 1.71 at $\delta = 10^{-5}$ (corresponding to $\sigma = 0.5$ and 1.08, respectively), Figure 2 shows the test

Table 1: Metrics for the final models for each configuration for each data set.

| Data set, metrics | Algorithm | Smaller $\sigma$ | Larger $\sigma$ |
|---|---|---|---|
| WikiText-2, test loss | DP-SGD cold | 5.2626 | 5.5627 |
| | DP-SGD warm | 5.1914 | 5.3288 |
| | PDA-DPMD | **5.1736** | **5.2956** |
| CIFAR-10, accuracy / test loss | DP-SGD cold | 62.9633 / 1.4225 | 40.6000 / 1.6890 |
| | DP-SGD warm | 66.3933 / 1.2371 | 53.4100 / 1.3462 |
| | PDA-DPMD | **67.0300 / 1.1435** | **55.3950 / 1.2785** |
| EMNIST, accuracy / test loss | DP-SGD cold | 87.5671 / 0.5422 | 84.7270 / 0.6170 |
| | DP-SGD warm | 87.8534 / 0.5089 | 86.3352 / 0.5586 |
| | PDA-DPMD | **87.9860 / 0.4706** | **86.7229 / 0.4982** |

loss for PDA-DPMD with $K = 100$ and $50$ respectively, and for the two baselines. From Table 1, we see that PDA-DPMD obtains the smallest test loss for both the privacy levels. Also, comparing the two DP-SGD baselines, we can see that using public data for pre-training provide trained models with higher utility.

Though our work's focus is on in-distribution public data, we additionally compare with SoTA (Asi et al., 2021) which uses WikiText-103 (Merity et al., 2017) as the public data. In that setting, our warm start DP-SGD baseline is better than the proposed SoTA in (Asi et al., 2021) by 1.1% for $\varepsilon = 1.0$ and 6.6% for $\varepsilon = 3.0$ in terms of test perplexity. See Appendix F for more details.

**Empirical Evaluation on CIFAR-10:** CIFAR-10 consists of 50,000 training images and 10,000 test images from 10 classes. After pre-training, we fine-tune the model with batch size 500 for 100 epochs. We search for optimal $K$ in $\{200, 500, 1000, 2000, 5000\}$. In Figure 3, for two different privacy levels, $\varepsilon = 3.51$ and $0.19$ at $\delta = 10^{-5}$ (corresponding to $\sigma = 1.51$ and $20.0$, respectively), we report the test loss and accuracy for $K = 2000$, and for the two baselines. From Table 1, we see that PDA-DPMD provides the best accuracy (even if by a small margin over the warm started DP-SGD baseline). Moreover, PDA-DPMD also results in significantly lower test loss compared to both the baselines for both privacy levels, which confirms with our theoretical analysis for the population loss.

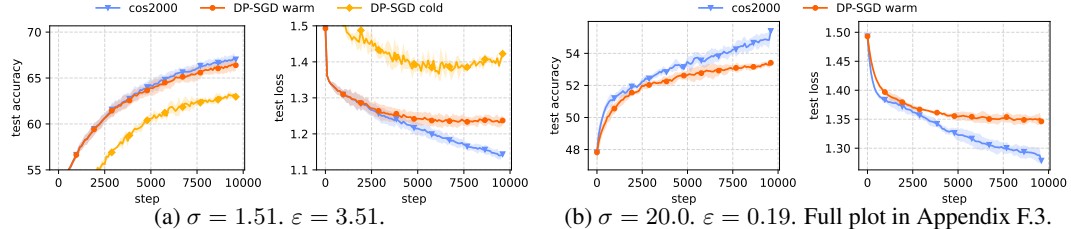

(a) $\sigma = 1.51$. $\varepsilon = 3.51$.      (b) $\sigma = 20.0$. $\varepsilon = 0.19$. Full plot in Appendix F.3.

Figure 3: CIFAR-10. 4% public data. Test accuracy / loss vs. training steps. Averaged over 3 runs.

**Empirical Evaluation on EMNIST:** EMNIST (ByMerge split) consists of 697,932 training images and 116,323 test images from 47 classes. After pre-training, we fine-tune with batch size 500 for 50 epochs. We search for optimal $K$ in $\{200, 500, 1000, 2000, 5000\}$. In Figure 4 and Table 1, for $\sigma = 0.41$ and $1.89$, corresponding to privacy $\varepsilon = 25.80$ and $0.48$ at $\delta = 10^{-6}$, we report the test loss and accuracy for $K = 500$, and for the two baselines. We see a similar trend as with CIFAR-10.

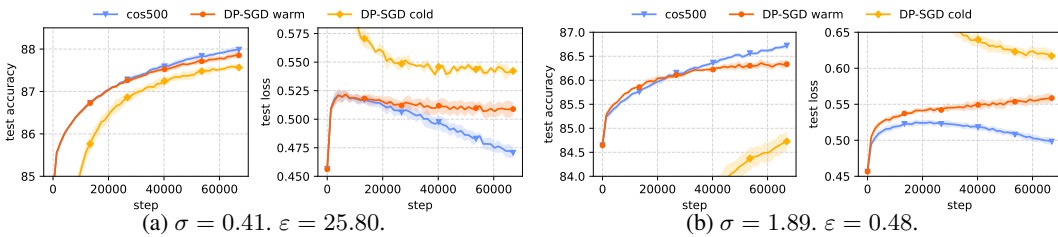

(a) $\sigma = 0.41$. $\varepsilon = 25.80$.      (b) $\sigma = 1.89$. $\varepsilon = 0.48$.

Figure 4: EMNIST. 4% public data. Test accuracy / loss vs. training steps. Averaged over 3 runs.

## 5 ETHICS AND REPRODUCIBILITY

**Ethics:** This work focuses on improving the privacy/utility trade-offs for model training with sensitivity data by using publicly available data. We envision that this work will make the adoption of (differentially) private model training more ubiquitous, and hence improve on the current privacy landscape in the context of machine learning. Our experimental results are on standard benchmark data publicly available data sets, and do not have any ethical concerns.

**Reproducibility:** Our experiments are either on synthetic data sets, or on standard publicly available data sets. We have provided full details (including hyperparameter choices) for the experimental setup to reproduce the results in the paper. We will make the code for our experiments public as soon as possible. Additionally, we have detailed proofs for all our theoretical results in the appendix.

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

# A NOTATION REFERENCE

We recall here the notation used throughout the paper:

| | |
|---|---|
| $\alpha$ | A strong convexity parameter |
| $\mathbf{b}$ | Noise added to gradients for privacy |
| $B_\Psi$ | The Bregman divergence induced by $\Psi$ |
| $\mathcal{C}$ | The constraint set |
| $d$ | A single sample |
| $D_{\mathsf{pub}}, D_{\mathsf{priv}}$ | The public and private data sets respectively |
| $\mathcal{D}$ | The universe of samples |
| $\varepsilon, \delta$ | The privacy parameters |
| $\eta$ | The learning rate/step size |
| $\mathbf{g}$ | A (batch) gradient |
| $G_Q$ | The Gaussian width of $Q$ |
| $\ell_{\mathsf{pub}}, \ell_{\mathsf{priv}}$ | The (per-example) public and private loss functions respectively |
| $L$ | The Lipschitz constant of $\ell_{\mathsf{priv}}$ |
| $\mathcal{L}$ | The empirical (private) loss on dataset $D$, i.e. $\mathcal{L}(\theta; D) = \frac{1}{\lvert D \rvert} \sum_{d \in D} \ell(\theta; d)$ |
| $n_{\mathsf{pub}}, n_{\mathsf{priv}}$ | The number of public and private samples respectively |
| $p$ | The dimensionality of the optimization problem |
| $\Psi$ | The empirical public loss, i.e. $\Psi(\theta) = \frac{1}{\lvert D_{\mathsf{pub}} \rvert} \sum_{d \in D_{\mathsf{pub}}} \ell(\theta; d)$; also the mirror map |
| $Q$ | A set such that $\Psi$ is 1-strongly convex with respect to $\lVert \cdot \rVert_Q$ |
| $T$ | The number of iterations in the optimization algorithm |
| $\theta$ | A solution to the optimization problem |
| $V$ | A bound on the variance of the public gradients |

# B MISSING BACKGROUND

**Mirror Maps:** A mirror map is a differentiable function $\Psi : \mathbb{R}^p \to \mathbb{R}$ that is strictly convex. Since $\Psi$ is strictly convex and differentiable, $\nabla\Psi : \mathbb{R}^p \to \mathbb{R}^p$ provides a bijection from $\mathbb{R}^p$ to itself. One can view $\theta$ as lying in a primal space and $\nabla\Psi(\theta)$ as lying in a dual space. In turn, we could now consider optimizing over the value $\nabla\Psi(\theta)$ in the dual space instead of $\theta$ primal space. Mirror descent does exactly that, performing gradient descent in the dual space by computing the gradient $\mathbf{g}_t = \nabla\ell(\theta_t)$ (where $\theta_t$ lies in the primal space), taking a step in the opposite direction in the dual space, and then using the inverse of the mirror map to determine $\theta_{t+1}$. Mirror descent is essentially motivated as minimizing a (linearized) loss plus a Bregman divergence (induced by $\Psi$) as the regularizer (Nemirovsky & Yudin, 1983). More formally, similar to proximal gradient descent, mirror descent is equivalent to taking the gradient $\mathbf{g}_t$ and performing the update $\theta_{t+1} = \arg\min_{\theta \in \mathcal{C}}[\eta \langle \mathbf{g}_t, \theta \rangle + B_\Psi(\theta, \theta_t)]$ where $B_\Psi(\theta_1, \theta_2) = \Psi(\theta_1) - \Psi(\theta_2) - \langle \nabla\Psi(\theta_2), \theta_1 - \theta_2 \rangle$ is the Bregman divergence generated by $\Psi$. Note that, if $\Psi(\theta) = \lVert \theta \rVert_2^2$, then the Bregman divergence is simply $B_\Psi(\theta_1, \theta_2) = \lVert \theta_1 - \theta_2 \rVert_2^2$ and mirror descent is equivalent to the usual gradient descent.

**Gaussian Width:** Given a bounded set $Q \subset \mathbb{R}^d$, the Gaussian width of $Q$, $G_Q$, is a measure of how isotropic the set is. $G_Q$ is defined as $\mathbb{E}_{g \sim N(0, \mathbb{I}_p)} \max_{x \in Q} \langle g, x \rangle$. Although the Gaussian width is well-defined for any bounded set, for this paper it suffices to consider defining the Gaussian width of convex sets containing the origin such that $\max_{x \in Q} \lVert x \rVert_2 = 1$. If $Q$ is just the unit $\ell_2$-ball, the "most isotropic" set satisfying this condition, then we have $G_Q = \sqrt{p}$; in particular, since every set $Q$ satisfying $\max_{x \in Q} \lVert x \rVert_2 = 1$ is contained in the $\ell_2$-ball, this is the maximum Gaussian width of any such set. On the other hand, if $Q$ is just the line from the origin to a single unit vector, we have $G_Q = \Theta(1)$. More generally, for any ellipsoid centered at the origin whose axes have radii $0 \le r_i \le 1, 1 \le i \le p$, we have that the Gaussian width of this ellipsoid is $\sqrt{\sum_{i=1}^p r_i^2}$. As other examples, the Gaussian width of the $\ell_1$-ball of radius 1 is roughly $\log p$, and the Gaussian width of the $\ell_\infty$ ball of radius $1/\sqrt{p}$ is roughly $\sqrt{p}$.

## C  MISSING PROOFS FROM SECTION 3

*Proof of Theorem 3.2.* The privacy guarantee follows from the moments accountant analysis of Abadi et al. (2016).

For the utility guarantee, following the analysis of Theorem 3.2 of Talwar et al. (2014), we have:

$$\mathbb{E}[\mathcal{L}(\theta_{\mathsf{priv}})] - \mathcal{L}(\theta^*) \leq \frac{B_\Psi(\theta^*, \theta_0)}{\eta T} + \eta \cdot O(L^2 \|Q\|_2^2 + \sigma^2(G_Q^2 + \|Q\|_2^2), \tag{2}$$

Let $\theta^*$ in particular be the minimizer satisfying Assumption 3.1. By $\alpha$-strong convexity, we have:

$$B_\Psi(\theta^*, \theta_0) = \Psi(\theta^*) - \Psi(\theta_0) - \nabla\Psi(\theta_0) \cdot (\theta^* - \theta_0) \leq \frac{1}{2\alpha} \|\nabla\Psi(\theta^*) - \nabla\Psi(\theta_0)\|_2^2.$$

Plugging this into Eq. (2) and noting that any $\Psi$ we sample is 1-strongly convex with respect to $\|\cdot\|_Q$, we get:

$$\mathbb{E}[\mathcal{L}(\theta_{\mathsf{priv}})] - \mathcal{L}(\theta^*) \leq \frac{\mathbb{E}\left[\|\nabla\Psi(\theta^*) - \nabla\Psi(\theta_0)\|_2^2\right]}{2\alpha\eta T} + \eta \cdot O(L^2 \|Q\|_2^2 + \sigma^2(G_Q^2 + \|Q\|_2^2)$$

We will show that without loss of generality, we can assume $\nabla\Psi(\theta_0) = \mathbf{0}$ and $\mathbb{E}_{d\sim\tau}[\nabla\ell_{\mathsf{pub}}(\theta^*; d)] = \mathbf{0}$. Once we have this assumption, Assumption 3.1 completes the proof.

The assumption follows since by convexity of $\mathcal{C}$ we have

$$\langle \nabla\Psi(\theta_0), \theta_0 - \theta^* \rangle \leq 0, \quad \langle \mathbb{E}_{d\sim\tau}[\nabla\ell_{\mathsf{pub}}(\theta^*; d)], \theta^* - \theta_0 \rangle \leq 0 \tag{3}$$

Then for any choice of $\Psi$ and $\mathcal{C}$ where either $\theta_0$ or $\theta^*$ is on the boundary of $\mathcal{C}$, suppose we extend $\mathcal{C}$ infinitesimally along the line $\{\theta_0 + c(\theta^* - \theta)|c \in \mathbb{R}\}$ (i.e., take a point on this line infinitesimally outside of $\mathcal{C}$ and update $\mathcal{C}$ to be the convex hull of itself and this point). Then by (3) we have that $\theta^*, \theta_0$, defined as the minimizers in $\mathcal{C}$, move apart from each other along this line and in turn by strong convexity the quantity $\|\nabla\Psi(\theta^*) - \nabla\Psi(\theta_0)\|_2^2$ cannot decrease. This implies that for any fixed $\ell_{\mathsf{pub}}$ and $\tau$, the quantity $\mathbb{E}\left[\|\nabla\Psi(\theta^*) - \nabla\Psi(\theta_0)\|_2^2\right]$ is maximized for a choice of $\mathcal{C}$ such that $\nabla\Psi(\theta_0) = \mathbf{0}$ and $\mathbb{E}_{d\sim\tau}[\nabla\ell_{\mathsf{pub}}(\theta^*; d)] = \mathbf{0}$. $\qquad\square$

*Proof of Lemma 3.3.* By convexity, for all $\theta \in \mathcal{C}$ we have $\mathcal{L}(\theta; D) \geq \mathcal{L}(\theta^*; D) + \langle \nabla\mathcal{L}(\theta^*; D), \theta - \theta^* \rangle$. Note that by optimality of $\theta^*$ and convexity, for all $\theta \in \mathcal{C}$ we have $\langle \mathbb{E}_{d\sim\tau}[\nabla\ell(\theta^*; d)], \theta - \theta^* \rangle \geq 0$. In turn, by the Cauchy-Schwarz inequality we can conclude that $\mathcal{L}(\theta^*; D) - \min_{\theta\in\mathcal{C}} \mathcal{L}(\theta; D)$ is always upper bounded by $\|\mathcal{C}\|_2 \cdot \|\nabla\mathcal{L}(\theta^*; D) - \mathbb{E}_{d\sim\tau}[\nabla\ell(\theta^*; d)]\|_2$. By $L$-Lipschitzness of each $\ell(\theta; d)$ we have:

$$\mathbb{E}_{D\sim\tau^n}\left[\|\nabla\mathcal{L}(\theta^*; D) - \mathbb{E}_{d\sim\tau}[\nabla\ell(\theta^*; d)]\|_2\right] \leq \frac{L}{\sqrt{n}},$$

Which completes the proof. $\qquad\square$

*Proof of Theorem 3.6.* Let $\mathbf{b}_t$ be the noise added for privacy. Without noise, mirror descent would set $\theta^*$ to be such that:

$$-\eta\mathbf{g}_t = \nabla\Psi(\theta^*) - \nabla\Psi(\theta_t).$$

Similarly, given the noisy gradient $\mathbf{g}_t + \mathbf{b}_t$, mirror descent would set $\hat{\theta}$ to be such that:

$$-\eta(\mathbf{g}_t + \mathbf{b}_t) = \nabla\Psi(\hat{\theta}) - \nabla\Psi(\theta_t).$$

We then have:

$$-\eta\mathbf{b}_t = \nabla\Psi(\hat{\theta}) - \nabla\Psi(\theta^*).$$

Since we assume the Hessian of $\Psi$ is defined everywhere, we have that $\nabla\Psi(\hat{\theta}) - \nabla\Psi(\theta^*) = \nabla^2\Psi(\widetilde{\theta})(\hat{\theta} - \theta^*)$ for some $\widetilde{\theta}$ on the line between $\hat{\theta}$ and $\theta^*$. In turn, we have:

$$\hat{\theta} - \theta^* = -\eta(\nabla^2\Psi(\widetilde{\theta}))^{-1}\mathbf{b}_t$$

The norm $x$ of $\mathbf{b}_t$ sampled from $N(0, \sigma^2 I_p)$ has the chi distribution, i.e. pdf proportional to $(x/\sigma)^{p-1}e^{-(x/\sigma)^2/2}$. In particular, this gives the following standard tail bound:

$$\mathbf{Pr}[\|X\|_2 > (1+c)\sqrt{p}\sigma] \leq \exp(-c^2 p/2).$$

So with probability at least $1 - e^{-c^2 p/2}$, we have the event $\mathcal{E}$ that $\|\mathbf{b}_t\|_2$ is at most $(1+c)\sqrt{p}\sigma$. Conditioned on this event, we have that $\hat{\theta}$, and thus $\widetilde{\theta}$, is in $E$, i.e. the lower bounds $\widetilde{w}_i$ apply to $\nabla^2\Psi(\widetilde{\theta})$. Since conditioning on $\mathcal{E}$ only decreases the expectation of $|\langle\hat{\theta} - \theta^*, \mathbf{v}\rangle|$, we have:

$$\mathbb{E}\left[\eta|\langle\hat{\theta} - \theta^*, \mathbf{v}\rangle||\mathcal{E}\right] \cdot \mathbf{Pr}[\mathcal{E}] = \eta\mathbb{E}\left[|\langle(\nabla^2\Psi(\widetilde{\theta}))^{-1}\mathbf{b}_t, \mathbf{v}\rangle||\mathcal{E}\right] \cdot \mathbf{Pr}[\mathcal{E}]$$

$$\leq \eta\mathbb{E}\left[|\langle(\sum_i \frac{1}{\widetilde{w}_i}\mathbf{v}_i\mathbf{v}_i^\top)\mathbf{b}_t, \mathbf{v}\rangle||\mathcal{E}\right]\cdot\mathbf{Pr}[\mathcal{E}] \leq \eta\mathbb{E}\left[|\langle(\sum_i \frac{1}{\widetilde{w}_i}\mathbf{v}_i\mathbf{v}_i^\top)\mathbf{b}_t, \mathbf{v}\rangle|\right] = \mathbb{E}\left[|\sum_i \frac{a_i}{\widetilde{w}_i}\langle\mathbf{b}_t, \mathbf{v}_i\rangle|\right]$$

$$= \eta\mathbb{E}\left[|\sum_i N(0, (a_i/\widetilde{w}_i)^2)|\right] = \eta\mathbb{E}\left[|N(0, \sum_i(a_i/\widetilde{w}_i)^2)|\right] = \sqrt{\frac{2}{\pi}}\cdot\eta\sigma\sqrt{\sum_i\left(\frac{a_i}{\widetilde{w}_i}\right)^2}.$$

When $\mathcal{E}$ does not happen, we have $w_i(\theta) \geq \alpha$ everywhere. So we have:

$$\mathbb{E}\left[\eta|\langle\hat{\theta} - \theta^*, v\rangle||\neg\mathcal{E}\right] \cdot \mathbf{Pr}[\neg\mathcal{E}] = \eta\mathbb{E}\left[|\langle(\nabla^2\Psi(\widetilde{\theta}))^{-1}b, v\rangle||\neg\mathcal{E}\right] \cdot \mathbf{Pr}[\neg\mathcal{E}]$$

$$\leq \eta\cdot\frac{1}{\alpha}\mathbb{E}\left[|\langle b, v\rangle||\neg\mathcal{E}\right]\cdot e^{-c^2 p/2}$$

To determine $\mathbb{E}\left[|\langle\mathbf{b}_t, \mathbf{v}\rangle||\neg\mathcal{E}\right]$, note that the distribution of $\langle\mathbf{b}_t, \mathbf{v}\rangle$ conditioned on $\neg\mathcal{E}$ is equivalent to the distribution of the first coordinate of $\mathbf{b}_t$ conditioned on $\neg\mathcal{E}$. We can sample $\mathbf{b}_t$ by first sampling its norm $\|\mathbf{b}_t\|_2$ conditioned on $\neg\mathcal{E}$, and then sampling a point on the sphere with radius $\|\mathbf{b}_t\|_2$ (no conditioning is required here). The expected absolute value of any coordinate $(\mathbf{b}_t)_i$ given $\|\mathbf{b}_t\|_2$ can be bounded as:

$$\mathbb{E}\left[|(\mathbf{b}_t)_i|\right] \leq \sqrt{\mathbb{E}\left[(\mathbf{b}_t)_i^2\right]} = \|\mathbf{b}_t\|_2/\sqrt{p}.$$

The inequality is Jensen's inequality, and the equality uses the fact that the coordinates $\mathbf{b}_i$ on the sphere are identically distributed, and so we have:

$$p\cdot\mathbb{E}\left[(\mathbf{b}_t)_i^2\right] = \mathbb{E}\left[\sum_i(\mathbf{b}_t)_i^2\right] = \|\mathbf{b}_t\|_2^2.$$

We now just need to bound the expectation of $\|\mathbf{b}_t\|_2$, given that it is at least $R$. Since the distribution of $\|\mathbf{b}_t\|_2/\sigma$ has pdf proportional to $x^{p-1}e^{-x^2/2}$, this expectation is $\sigma$ times:

$$\frac{\int_{(1+c)\sqrt{p}}^\infty x^p e^{-x^2/2}}{\int_{(1+c)\sqrt{p}}^\infty x^{p-1}e^{-x^2/2}} = \frac{\Gamma((p+1)/2)(1 - P((p+1)/2, (1+c)^2 p/2))}{\sqrt{2}\Gamma(p/2)(1 - P(p/2, (1+c)^2 p/2))}.$$

Where $\Gamma$ is the gamma function and $P$ is the regularized gamma function. Analytically, we can verify that $\frac{\Gamma((p+1)/2)}{\Gamma(p/2)} \leq \sqrt{p/2}$ for all $p \geq 1$, and $\frac{(1-P((p+1)/2,(1+c)^2 p/2))}{(1-P(p/2,(1+c)^2 p/2))} \leq 3(1+c)^2$ for all $p \geq 1$. So we get:

$$\mathbb{E}\left[\|\mathbf{b}_t\|_2 \,|\neg\mathcal{E}\right] \leq \frac{3(1+c)^2}{2}\sqrt{p}\sigma$$

Putting it all together, we get:

$$\mathbb{E}\left[\eta|\langle\hat{\theta}-\theta^*, v\rangle||\neg\mathcal{E}\right] \cdot \mathbf{Pr}[\neg\mathcal{E}] \leq \eta \cdot \frac{1}{\alpha} \cdot \frac{3(1+c)^2}{2} \cdot e^{-c^2 p/2} \cdot \sigma$$

Now applying the law of total expectation gives the theorem statement. $\qquad\square$

## D  EXAMPLES OF STOCHASTIC CONVEX OPTIMIZATION PROBLEMS

In this section, we consider two canonical stochastic convex optimization problems and the associated quantity $G_Q$, to help understand the bound in Theorem 3.5. In both cases we specify the public loss; as previously mentioned, a natural choice is to take the private loss to be the clipped version of the public loss to ensure Lipschitzness holds.

**Mean Estimation:** Suppose we assume we know the true covariance matrix $\Sigma$ in the mean estimation problem for data drawn from a multivariate Gaussian. Then mean estimation is equivalent to minimizing loss function $\frac{1}{2n}\sum_i (x_i-\theta)^\top \Sigma^{-1}(x_i-\theta)$, where $x_i$ is the $i$th sample point, and $\theta$ is our estimated mean. The Hessian is $\Sigma^{-1}$. By rescaling, wlog assume the maximum eigenvalue of $\Sigma$ is 1. Now, the smallest set $Q$ such that the public loss function is 1-strongly convex with respect to the $Q$-norm is the ellipsoid centered at the origin whose axes are in the directions of the eigenvectors of $\Sigma$, and whose length in the direction of the eigenvector $v_i$ is the corresponding eigenvalue $\lambda_i$. The Gaussian width of $Q$ is thus $\sqrt{\sum_{i=1}^p \lambda_i^2}$. In particular, if e.g. $\Sigma$ only has a few large eigenvalues (i.e., the data only has variance in a few directions), this is much smaller than $p$.

**Linear Regression:** We choose the loss to be the mean squared error. Now, the public loss function is proportional to $\|\mathbf{X}\theta - \mathbf{y}\|_2^2$, where $\mathbf{X}$ is the (public) feature matrix, $\mathbf{y}$ are the dependent variables, and $\theta$ is our model. In particular, this has Hessian $\mathbf{X}^\top\mathbf{X}$. By normalizing, we can assume wlog the smallest eigenvalue of this matrix is 1, and so similarly to mean estimation the quantity $G_Q$ is $\sqrt{\sum_{i=1}^p 1/\lambda_i^2}$, where $\lambda_i$ are the eigenvalues of $\mathbf{X}^\top\mathbf{X}$ (i.e., the singular values of $\mathbf{X}$). Note that unlike in the previous example (where we assumed we knew $\Sigma$ a priori and thus the Hessian was fixed), the Hessian here depends on the public data.

## E  ADDITIONAL DETAILS FOR THE EXPERIMENT ON LINEAR REGRESSION

Note that the exact optimum on the public data can be computed exactly as $\theta^*_{\text{pub}} = (\mathbf{X}^\top\mathbf{X})^{-1}\mathbf{X}^\top\mathbf{y}$. The mirror descent step can also be solved exactly by applying the inverse of the Hessian $\mathbf{X}^\top\mathbf{X}$ to the gradient, since the Hessian is the same everywhere.

For numerical stability, we add a small constant times the identity matrix to the Hessian before computing its inverse. We also normalize the Hessian of the loss function so its inverse (which is applied to the gradient before taking a step in PDA-DPMD) has maximum eigenvalue of one. This ensures that if the Hessian were a multiple of the identity matrix, DP-SGD and PDA-DPMD would behave exactly the same for the same hyperparameter choice.

We perform a grid search over the learning rate, clipping norm, and number of epochs used and report the best empirical loss. We perform 20 trials for each algorithm and dimension.

# F  ADDITIONAL DETAILS ON REAL-WORLD EXPERIMENTS

## F.1  FIRST-ORDER APPROXIMATION TO MIRROR DESCENT

In practice, the Mirror Descent (MD) step in Line 4 of Algorithm 1 is computationally the most challenging. In the following, we provide an approximation of this step in the setting where i) the problem is *unconstrained*, i.e., $\mathcal{C} = \mathbb{R}^p$ and ii). the public loss $\Psi(\theta)$ may not be strongly convex with respect to the $\ell_2$-norm. This approximation makes Algorithm 1 efficient in practice.

Consider the following equivalence of Line 4 of Algorithm 1, with $\Psi(\theta)$, the public loss on $D_{\mathsf{pub}}$, replaced by $\widehat{\Psi}(\theta) = \Psi(\theta) + \frac{1}{2}\|\theta\|_2^2$. This follows from (Hazan, 2019, Lemma 5.5).

$$\theta_{t+1} \leftarrow \underset{\theta \in \mathbb{R}^p}{\arg\min} \left\langle \sum_{i=1}^{t} \eta_i\left(\mathbf{g}_i + \mathbf{b}_i\right), \theta \right\rangle + \widehat{\Psi}(\theta) \tag{4}$$

$$= \underset{\theta \in \mathbb{R}^p}{\arg\min} \sum_{i=1}^{t} \eta_i \left( \langle \mathbf{g}_i + \mathbf{b}_i, \theta \rangle + \frac{1}{t\eta_i}\Psi(\theta) \right) + \frac{1}{2}\|\theta\|_2^2$$

$$\approx \underset{\theta \in \mathbb{R}^p}{\arg\min} \sum_{i=1}^{t} \eta_i \left\langle \mathbf{g}_i + \mathbf{b}_i + \frac{1}{t\eta_i}\nabla\Psi(\theta_i), \theta \right\rangle + \frac{1}{2}\|\theta\|_2^2, \tag{5}$$

where (5) follows from the first-order approximation $\Psi(\theta) \approx \Psi(\theta_i) + \langle \nabla\Psi(\theta_i), \theta - \theta_i \rangle$.

In the experiments, we replace $\mathbf{g}_i + \mathbf{b}_i + \frac{1}{t \cdot \eta_i}\nabla\Psi(\theta_i)$ with $\alpha_i(\mathbf{g}_i + \mathbf{b}_i) + (1 - \alpha_i)\nabla\Psi(\theta_i)$, where $\alpha_i \in (0, 1]$. This reparamertization helps with more effective hyperparameter tuning while training deep learning models. We therefore have the update rule (1).

## F.2  SETUP

**Network architectures:** Table 2 shows model architectures for CIFAR-10, EMNIST, & WikiText-2.

Table 2: Model architectures for real data experiments.

(a) Model architecture for CIFAR-10.

| Layer | Parameters |
|---|---|
| Convolution $\times 2$ | 32 filters of $3 \times 3$, strides 1 |
| Max-Pooling | $2 \times 2$, stride 2 |
| Convolution $\times 2$ | 64 filters of $3 \times 3$, strides 1 |
| Max-Pooling | $2 \times 2$, stride 2 |
| Convolution $\times 2$ | 128 filters of $3 \times 3$, strides 1 |
| Max-Pooling | $2 \times 2$, stride 2 |
| Fully connected | 128 units |
| Softmax | - |

(b) Model architecture for EMNIST.

| Layer | Parameters |
|---|---|
| Convolution | 16 filters of $8 \times 8$, strides 2 |
| Convolution | 32 filters of $4 \times 4$, strides 2 |
| Fully connected | 32 units |
| Softmax | - |

(c) Model architecture for WikiText-2.

| Layer | Parameters |
|---|---|
| Input | 8000 |
| Fully connected | 120 |
| LSTM $\times 2$ | 120 hidden units |
| Fully connected | 8000 |
| Softmax | - |

**Hyperparameter Tuning:** We keep the clipping norm to be 1. One small difference with the standard DP-SGD update rule is that we enforce an additional clipping step for the privatized gradient for the image classification task, where the clipping norm is the same as the clipping norm of for

individual gradient. The reason for this additional step is that the norm of the averaged clipping gradients should still be upper bounded by the clipping norm.

The only hyperparameters that need to be tuned are the learning rate and $K$ that controls the decaying of $\alpha_t$. For the learning rate, we consider a grid of $\{1, 2, 5\} \times 10^i$ for different $i$s such that the optimal learning rate does not appear on the boundary. We search for the optimal $K$ in $\{100, 200, 500\}$ for WikiText-2 and $\{200, 500, 1000, 2000, 5000\}$ for the image classification tasks.

## F.3 FULL PLOTS FOR SECTION 4

In Figure 5, we plot the complete version of Figure 2, and in Figure 6, we show the complete version of Figure 3.

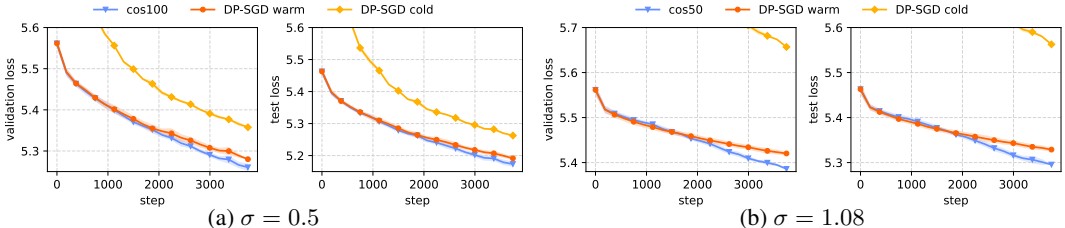

Figure 5: Full plot for Figure 2. WikiText-2. 4% public data.

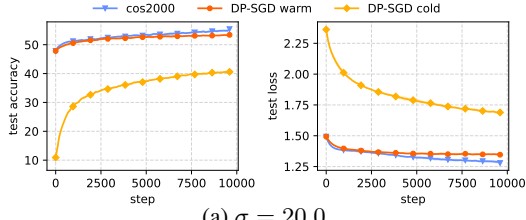

Figure 6: Full plot of Figure 3. CIFAR-10. 4% public data. Test accuracy / loss vs. training steps. Averaged over 3 runs.

## F.4 WIKITEXT-2 WITH WIKITEXT-103 AS PUBLIC DATA

We compare with the SoTA (Asi et al., 2021) which uses WikiText-103 as public data. Specifically, we consider their "LargeAux" setting under $\varepsilon = 1.0$ and $3.0$. Since the implementation for Asi et al. (2021) is not public as of writing this work, we make our best effort to match their experiment setup. We note that the data preprocessing and the number of iterations used (thus the noise multiplier for achieving the same $\varepsilon$) might differ.

We preprocess WikiText-103 as follows. After processing WikiText-2 as described in Section 4, we convert all words that does not appear in the processed WikiText-2 as the unknown token. Then, we split the sentences into length-35 sequences, and remove all sequences that overlap with WikiText-2. Finally, we randomly sample 48,764 sequences, in order to match the "LargeAux" setting where the public dataset is of the same size as the private training dataset.

Figure 7 shows the results. In our setting, cold start DP-SGD reaches similar log perplexity as those in (Asi et al., 2021), while the warm-start DP-SGD is already better than Asi et al. (2021) (LargeAux). The final test log perplexities are summarized below, with the results in (Asi et al., 2021) converted from perplexity to log perplexity.

| Algorithm | $\varepsilon = 3.0$ | $\varepsilon = 1.0$ |
|---|---|---|
| Asi et al. (2021) DP-SGD (cold) | 5.4819 | 5.6623 |
| Asi et al. (2021) (LargeAux) | 5.4324 | 5.5254 |
| Our DP-SGD (cold) | 5.4030 | 5.5956 |
| Our DP-SGD (warm) | 5.3646 | 5.5141 |

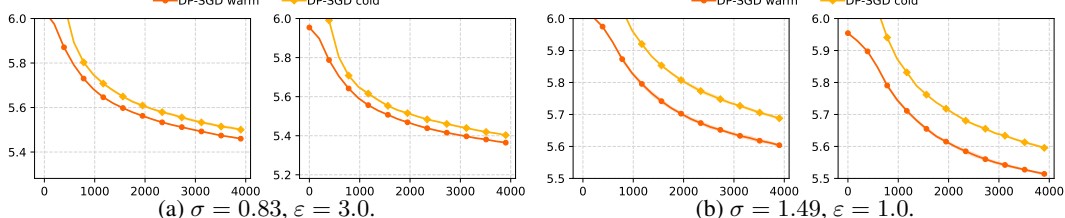

(a) $\sigma = 0.83$, $\varepsilon = 3.0$.  (b) $\sigma = 1.49$, $\varepsilon = 1.0$.

Figure 7: WikiText-2. WikiText-103 as public data.

