# OpenReview forum: "Public Data-Assisted Mirror Descent for Private Model Training"
_ICLR.cc/2022/Conference — ICLR 2022 Submitted_

### Official Review · Reviewer_mr6n · 2021-11-02

**Correctness:** 4
**Technical Novelty And Significance:** 3
**Empirical Novelty And Significance:** 3
**Recommendation:** 5
**Confidence:** 5

**Main Review:**

Strengths:
1. The authors use public data as the mirror map to provide improved bounds compared with the SOTA. This idea is quite interesting and may be used to other problems.
2. They further show that their algorithm has a natural "noise stability" property.
3. Although DP Mirror Descent has been studied before, they were no experimental result. This paper provide some experimental results.

Weakness:
1. Compared with the classical case ("Private empirical risk minimization beyond the
worst case: The effect of the constraint set geometry") The improvement on the upper bound is limited if there is not enough data.

2. To achieve the upper bound, the running time needs to be $O(n^3)$, which is lower than the SOTA $O(n^{1.5})$  [1].

3. The paper did not compared with the SOTA ("Private empirical risk minimization beyond the
worst case: The effect of the constraint set geometry") in experiments. I think this is necessary as this is the most close method, instead of DP-SGD.

Moreover, the authors left the reference [1], which also studied the problem.
[1]Wang, Di, Minwei Ye, and Jinhui Xu. "Differentially private empirical risk minimization revisited: Faster and more general." arXiv preprint arXiv:1802.05251 (2018).

**Summary Of The Paper:**

In this paper, the authors study differentially private empirical risk minimization (DP-ERM). Specifically, they study the case where the constraint set $\mathcal{C}$ has additional geometric structure, i.e., its Gaussian width could much lower than the underlying dimension $p$, such as the $\ell_1$-norm ball. The paper has been studied previously. However, this paper assume there are some additional public data. Specifically, they apply Mirror Descent with the loss generated by the public data as the *mirror map*, and using DP gradients of the loss generated by the private (sensitive) data. They also provide some experiments to show the performance of their algorithm.

**Summary Of The Review:**

As I mentioned, due to high time complexity (which is impractical), lack of comparison with SOTA in experiments and limited improvement on the upper bound. I tend to reject the paper.
--------
After the rebuttal, the reviewer solved mots of my concerns instead of the time complexity. So I raise the score to 5.

---

> ### Author Response · Authors · 2021-11-14
> **Response to reviewer mr6n**
>
> We believe that there may have been some confusion w.r.t. the main contributions of the paper, especially in the context of Wang et al. and Talwar et al. In the following, we provide clarifications for those:
>
> 1. While our result gets benefit if the constraint set $\mathcal{C}$ has smaller gaussian width than the $\ell_2$-ball, we still get dimension independent excess empirical risk, and population risk with only $p$ public examples, even when $\mathcal{C}$ is the $\ell_2$-ball. It is also known that without access to public data such a result is impossible to achieve because of the lower bounds in Bassily et al. 2014 (https://arxiv.org/abs/1405.7085). Both the prior works mentioned in the review do have a necessary $\sqrt{p}$ dependence on the excess risk, when the constraint set is the $\ell_2$-ball.
>
> 2. While Talwar et al. (i.e., the Private ERM beyond the worst-case:...) study a differentially private variant of mirror descent, the potential/divergence function they consider is a fixed function based on the geometry of the constraint set $\mathcal{C}$, rather than something dependent on some public data set (which is what we do). In particular, the result of Talwar et al. can be thought of a special case of ours, when we ignore the public data, and use a fixed function as the potential/divergence function. For reasons mentioned in bullet (1) above, provably, it is not possible for Talwar et al. to achieve the utility/privacy trade-offs we achieve. As a result it may not be fair comparing Talwar et al. to ours as the current SoTA, as it does not have access to public data. In addition, it’s not even clear what choice of $\Psi$ one would use for the algorithm of Talwar et al. in our experiments, as their algorithm chooses $\Psi$ based on the constraint set, but in our experiments we perform unconstrained optimization. We will add this discussion to the comparison section of the paper to make it more clear.
>
> 3. We are not sure how the time complexity bound for our paper (n^3) was derived. In full generality our algorithm requires solving a Bregman divergence penalized optimization problem at every iteration of the training process. Any form of acceleration on the standard mirror descent (similar to that in [1]) is also directly compatible with our work. It is worth noting that in the empirical section (Section 4, and equation (1)) we do provide an approximate variant of the mirror descent optimization that has an oracle complexity (in regards to number of gradient evaluations) to be T (number of training steps) * (|public minibatch size|+|private minibatch size|). For all the empirical settings considered in the paper, this number is much smaller than n^{1.5} (mentioned in the review).

---

### Official Review · Reviewer_cMgL · 2021-11-03

**Correctness:** 4
**Technical Novelty And Significance:** 3
**Empirical Novelty And Significance:** 4
**Recommendation:** 6
**Confidence:** 4

**Main Review:**

Strengths:
1. This paper proposes a new idea to use public data implicitly by using the public data loss as the mirror map in DP mirror descent.
2. The empirical results of this paper on real world datasets seem to beat current baselines.

Questions/Clarifications:

1. In my opinion, this paper needs better writing. At many places, mathematical notation is either used but not defined explicitly or sometimes not consistent. A short notation section, would make its easier to read. I will point out a few places where I found it particularly confusing.

    a. I couldn't find the definition of $||\cdot||_Q$, although it maybe standard, its better to explicitly define it to clear any confusion.
    b. In lemma 3.3, how is $\ell(\theta;D)$ defined? It doesn't parse with the definition of $\ell$ in the first line.
    c. Usually $\mathcal{L}$ denotes the population loss and $\hat{\mathcal{L}}$ denotes the empirical loss, this notation is overloaded and causes confusion


2. The way to bound the excess population risk seems non-standard. If I understand correctly, first the difference of empirical loss of the estimate and the population optimum is bounded, lemma 3.3 is supposed to relate the population optimum to the sample (empirical) optimum, and then Lemma 3.4 is used to relate the excess population risk and excess empirical loss. Is there some intuition as to why this path was chosen? Why can one not directly either give an algorithm with population loss bound or give an excess empirical loss bound and add the generalization error?

3. One of the biggest causes of concern for me is the Assumption 3.1, it assumes that some minimizer of the private loss is also the (unique) minimizer of the public loss. Can the authors give a real world example where the two losses are not the same but this holds?

4. Overall, the paper lacks toy examples which would help understand the theory better and clear doubts that arise when reading assumptions. Some simple toy examples could be linear regression, logistic regression, LASSO, etc. I'd be curious to see what are the choices of private and public loss for these cases, what is the Gaussian width and what does the guarantee look like for these values?

5. Since the minimizers of the public loss and one of the minimizers of the private loss is the same, one trivial baseline to compare against is doing non-private training using only public data. It seems the setting here is that of public data being a constant but small fraction of the private data. What is the comparison of errors for this setting?

6. In the proof of Theorem 3.2, if $\nabla \Phi(\theta_0) = 0$, since $\Phi$ is the public loss, which is strongly convex, this would imply $\theta_0$ needs to be the empirical public loss minimizer? It would be better if a formal argument is made for this.

**Summary Of The Paper:**

In this paper, a new algorithm has been proposed which leverages in-distribution public data to provide improvements in private training. The algorithm uses the loss on public data (with a strongly convex loss function) as a “mirror map” to implement private mirror descent on the private data. It is shown to give dimension independent bounds in certain regimes. Empirically, on both synthetic and real world datasets it is shown to perform favourably compared to recent work.

**Summary Of The Review:**

There are some issues with the writing and some questions regarding the validity of assumptions and the validity the claim of dimension independence in usual settings, as detailed in the main review. Thus my score of 5.

---

> ### Author Response · Authors · 2021-11-14
> **Response to reviewer cMgL**
>
> We are thankful to the reviewer for the insightful suggestions on improving the paper. We address the questions/clarifications of the reviewer here, as well as any changes we plan to make to the paper based on the reviewer’s feedback. Due to the character limit, our response is split into two comments since we wish to meaningfully address all the reviewer's questions and concerns.
>
> 1. We will add a notation section in the updated submission as suggested by the reviewer, and ensure the entirety of the paper is consistent with that notation section. The Q-norm/Minkowski norm (defined as $\min\{c \in \mathbb{R}_{\geq 0} : x \in cQ\}$) is now directly defined in the statement of Theorem 3.2. The notation used in Lemma 3.3 and its proof is now consistent with Section 1.1.
>
> 2. The intuition for why we obtain the empirical excess loss bound by first comparing to the population optimizer and then translating to a bound comparing to the empirical optimizer is explained in the paragraph titled “Dimension Independent Empirical Risk Bounds for Convex Losses”, requoted here for convenience:
>
> “We bound the excess empirical loss  … for this reason.”
>
> To elaborate: The excess empirical loss bound compared to a point $\theta$ that we derive is a function of $\nabla \Psi(\theta)$. For $\theta = \theta^*$, i.e. the population minimizer, the bounded variance assumption tells us that $\nabla \Psi(\theta)$ is small in expectation, which gives us a good bound on the empirical loss. Suppose instead we chose $\theta$ to be the empirical minimizer of the private loss function, which as the reviewer points out is the standard way to derive population loss bounds. Since the private loss function is only assumed to be convex, for some choices of $\tau$ (the distribution of samples) and $\ell_{priv}$ the empirical minimizer will be distance $diam(\mathcal{C})$ from the population minimizer with constant probability (over the choice of $D_{priv}$). In particular, it could be far in a direction where $\nabla \Psi$ grows very quickly. In turn, we cannot hope to reasonably bound $\nabla \Psi(\theta)$ where $\theta$ is the empirical minimizer of the private loss, and so our empirical loss bound using this approach could be quite poor.
>
> 3.  We would first like to emphasize the motivation for the problem setup in the paper using two separate loss functions: to allow a more generalized result where the public loss can be strictly convex (as mirror descent is ill-defined if $\Psi$ is convex but not strictly convex), while the private loss is Lipschitz (which is standard/often necessary for theoretical results on private optimization). We mention here two practical settings captured by the problem setup, where the minimizer could be shared.
>
> As mentioned in the last paragraph of section 1.1, we expect that in practice one will choose the public and private loss function to be the same (in which case they would have the same minimizer if the public data is in-distribution). However, in practice when optimizing using the private samples, we will clip their gradients, so from the optimization perspective $\ell_{priv}$ is not actually the same as $\ell_{pub}$, but instead the function whose gradients are the clipped gradients of $\ell_{pub}$.
>
> In turn, if clipping does not affect the minimizer, we have a practical scenario where from the optimization perspective the loss functions differ, but the minimizers are the same. For example, if we perform e.g. linear regression with mean squared error as the loss, and the error is drawn from a symmetric distribution, then clipping should not affect the population minimizer.
>
> Another possibility is that we again choose the public and private loss function to be the same, and the loss function is already Lipschitz, so clipping is not necessary. But in this case, $\nabla \Psi$ is not a bijection from $\mathbb{R}^p$ to itself, so mirror descent is ill-defined. A simple fix is to add an $\ell_2$-regularizer to the public loss function to ensure it satisfies strict/strong convexity. Now, since mirror descent only cares about the differences in the gradients of $\Psi$ rather than the value of $\nabla \Psi$ evaluated at any specific point, where the $\ell_2$-regularizer is centered will not affect the behavior of mirror descent, so we can analyze the algorithm as if the regularizer is centered such that the public and private loss functions still share a minimizer.

---

> > ### Author Response · Authors · 2021-11-14
> > **Response to reviewer cMgL (pt. 2)**
> >
> >
> >
> > 4. We thank the reviewer for suggesting adding toy examples to help understand some of the parameters in our empirical loss bound. We will add a section to the paper to do precisely this.  We can give one such example here: Consider linear regression. Here, we would choose the public loss to be the mean squared error, and the private loss to be the clipped version of the mean squared error. Now, the public loss function is $\lVert Ax-b\rVert_2^2$, where $A$ is the feature matrix, $b$ are the dependent variables, and $x$ is our model. In particular, this has Hessian $A^T A$. By normalizing, we can assume wlog the smallest eigenvalue of this matrix is 1. Now, the smallest set $Q$ such that the public loss function is 1-strongly convex with respect to the $Q$-norm is the ellipsoid centered at the origin whose axes are in the directions of the eigenvectors of $A^T A$, and whose length in the direction of the eigenvector $v_i$ is 1/$\lambda_i$. The Gaussian width of $Q$ is $\sqrt{\sum_{i=1}^p 1/\lambda_i^2}$. Since we assume $\lambda_i \geq 1$ for all $i$, this is at most $\sqrt{p}$, but can be much smaller if $A^T A$ has many large eigenvalues. The variance $V$ in our empirical loss bound is then determined by the distribution of feature vectors. For example, if, as in our synthetic data experiments, the feature vectors are sparse, then we would get $V = O(1)$.
> >
> > 5. If we consider also using the public data to get low excess population loss on $\ell_{priv}$, with $n_{pub}$ public samples only, our excess population loss will decay as $1/\sqrt{n_{pub}}$. We will mention this explicitly in the paper in our theoretical section for a fair comparison to our bound. Note that if, say, $n_{pub} \approx n_{priv}$, then in terms of getting good asymptotic bounds, the private data does not help regardless of what optimization method we use. So it is only interesting to compare this to our bounds when n_{pub} << n_{priv}. In particular, if $n_{pub} = G_Q \leq \sqrt{p} \leq n_{priv}$ (results in the DP SCO literature often assume $\sqrt{p} \leq n_{priv}$), $1/\sqrt{n_{pub}}$ is asymptotically worse than our population loss bound.
> >
> > It is also worth noting that in our deep learning experiments, we ran both PDA-DPMD and DP-SGD from a “warm-start” on the public data, and in all these experiments training on the private data gave substantial improvements in accuracy over the warm-start even after only a few epochs of training. Furthermore, in many of the experiments training on the private data from a cold-start still achieved higher accuracy than training on public data. So in the settings we consider, it appears training using only the public data is a much easier baseline to beat than the DP-SGD baselines we considered in our empirical sections.
> >
> > 6. As defined in Line 1 of Algorithm 1, we choose $\theta_0$ to be the minimizer of the empirical public loss. This corresponds to pre-training on the public data for a warm start, as we did in our experiments.
> >
> > To clarify what the proof is doing: the proof of Theorem 3.2 is effectively arguing that the norm of $\nabla \Psi(\theta_0) - \nabla \Psi(\theta^*)$ is maximized when the constraint set is $\mathbb{R}^p$ (informally, this is true because if we have a smaller constraint set, expanding the constraint set can only bring $\theta_0$ and $\theta^*$ further apart, and since $\Psi$ is strongly convex this increases the difference between their gradients). In this case, since $\Psi$ is strongly convex, we can assume $\nabla \Psi(\theta_0) = 0$ since $\theta_0$ is the minimizer of $\Psi$.

---

> > > ### Comment · Reviewer_cMgL · 2021-11-20
> > > **Some more questions**
> > >
> > > Thank you for your response, it is very informative.
> > >
> > > 1. I think the clipping based example could be a good canonical example for this setting, but I have a few questions regarding that. Do you have a reference as to what the gradient clipped version of a function looks like? Is it only defined in an implicit form? What properties of the old function whose gradients are clipped does it still satisfy? Is it still convex if the old function is convex?
> > >
> > > 2. The second example with the regularizer isn't very convincing for me since it seems like you might need to know too much about the function (effectively something strong about the set of minimizers?) to correctly center it.
> > >
> > > 3. Based on the second last paragraph before the contributions section, I presumed we were in the setting of public data being a constant fraction (although small fraction) of the private data. Is the setting you consider one in which the public data is lower order than the private data?
> > >
> > > 4. What is the reason that the excess empirical and population risk considered is that on the private loss? Based on the clipping example say for linear regression, wouldn't it make more sense to consider the excess population and empirical risk defined using the public loss? In that case, I am not sure of the excess risk results of PDA-DPMD, but training on just $\ell_{pub}$ would give $1/n_{pub}$ error since it is strongly convex.
> > >
> > > There might still be settings where the results are interesting. I am trying to understand what are those settings.

---

> > > > ### Author Response · Authors · 2021-11-21
> > > > **Responses to additional questions**
> > > >
> > > > Thank you for the response! We attempt to answer your followups here:
> > > > 1. The clipped function may not be necessarily convex. However, there are settings where it is indeed convex, and can be explicitly defined. For example, in the case of generalized linear models it is in fact the huberized loss. (See Section 5.1 in http://proceedings.mlr.press/v130/song21a/song21a.pdf for a formal description).
> > > > 2. For the mirror descent steps in our algorithm we only need to know the differences between gradients, which for e.g. the $\ell_2^2$ regularizer would not depend on where it is centered. So practically one can implement mirror descent using the public loss as the mirror map without actually choosing where to center the regularizer.
> > > > We would like to remind the reviewer that we are using these assumptions for the theoretical utility proof only. Additionally, even in the most natural setting where the public loss function is equal to the private loss function (this can be the case while maintaining strong convexity of the public loss if e.g. the constraint set C has a small enough radius such that clipping is not needed), we still provide dimension independent excess empirical risk with only n_pub = p samples (even in the constrained optimization setting) which was not known earlier without relying on convexity for privacy. So, in some sense the generality of the assumption (where the private and public losses can differ in arbitrary ways as long as the assumption holds) was meant to capture a broader class of scenarios, but was not intended to be restrictive.
> > > > 3. We would like to apologize for the confusion. For the theoretical results the regime where our results are interesting is when the public data (n_pub) is of the order of the number of dimensions in the model (p), which in turn is to be O((n_priv)^{2/3}). If the number of public samples is indeed a constant fraction of the number of private samples, in terms of asymptotic bounds one can just ignore the public examples, so unless we are optimizing constants this setting is not theoretically interesting. The mention of the public data being a small fraction of the private data samples is only for the deep learning empirical results. There we do show that with the public data alone it is not possible to reach the accuracy we achieve with PDA-DPMD in the deep learning experiments (the result of training only on public data is the starting point on our loss/accuracy curves, which is improved on by the private training). We note that our synthetic linear regression experimental results more closely reflect the regime considered in the theoretical results, where the number of public examples scales with the dimension but can be substantially smaller than the number of private examples.
> > > > 4. We chose to present our excess loss bounds for the private loss that is convex-Lipschitz, since convex-Lipschitz losses are the standard assumption in the theoretical private SCO literature, and so by considering this setting our results are more directly comparable to past theoretical work. It is a natural question to ask if it makes sense to bound the excess loss on the clipped function rather than the unclipped function, but it is known that unless the loss functions are convex GLMs, clipping may result in vector fields that may not conform to the gradient field of a fixed function (see Theorem 5.3 inhttps://arxiv.org/abs/2006.06783). Hence we believe the private SCO literature primarily focuses on convex Lipschitz functions.
> > > > In addition, training on n_{pub} public samples will indeed obtain an excess empirical risk of 1/n_{pub} on the public loss but the setting we are interested in for theoretical purposes is the one mentioned for response (1), where just training on public data alone will result in an asymptotically much worse error.

---

> > > > > ### Comment · Reviewer_cMgL · 2021-11-22
> > > > > **Thank you for your response**
> > > > >
> > > > > Thank you for your response! I would strongly recommend adding the examples of clipping and regularizer in the main body. As mentioned in point 2 of the previous response, currently the assumption seems to be more restrictive than trying to capture a broader class of scenarios. I would also recommend citing http://proceedings.mlr.press/v130/song21a/song21a.pdf to strengthen the examples you provide.
> > > > >
> > > > > My questions on the theory have been resolved and keeping in mind the empirical contributions of the paper, I will update my score and assessment.

---

> > > > > > ### Author Response · Authors · 2021-11-22
> > > > > > **New revision**
> > > > > >
> > > > > > Thanks for the feedback and updated review! We are glad that we were able to resolve your questions on the theoretical side. We have just uploaded a second revision where we added a discussion of the scenarios brought up in this comment chain under Assumption 3.1, and include a citation to the Song et al. paper discussion clipping.

---

### Official Review · Reviewer_GaSe · 2021-11-03

**Correctness:** 4
**Technical Novelty And Significance:** 4
**Empirical Novelty And Significance:** 4
**Recommendation:** 8
**Confidence:** 4

**Main Review:**

This is a beautiful paper! It is well written, easy to follow, covers the literature very well. The results are nice and for me and the approach very natural. It is an easy accept.

I have few comments about the paper though. Since the paper works in the public-private data setting, it would be nice to know (even theoretically) the comparison between how much public data is used by previous work and we have access to equivalent data, what is the performance of the current submission. Likewise, if we have a given utility to aim for, what is the difference between the number of public data samples in the current submission vs previous works. This would put the work in a very precise light as to whether it is improving the state-of-the-art or not.

The paper mentions in the appendix that they compare SoTA with Asi et al. and they made a best effort to match their experiment set up. Asi et al. is a published work in ICML 2021 and I am surprised by the claim of the authors that the code is not publicly available -- given that ICML requires code submission in the supplementary material. If it is indeed so, I would suggest the authors to reach out to the authors of Asi et al. to get their code because the hyperparameters used in this submission can be very different from what Asi et al. did and hence the claim may or may not be true.

One question I have to the authors is also with regards to some other works that came out in different geometry (Bassily et al. (COLT 21), Asi, Feldman, Koren, and Talwar ICML 21 (AFKT to differentiate from Asi et al. mentioned in this paper), and Kulkarni et al. (https://arxiv.org/abs/2103.15352). Granted they do not consider public-private data setting, at least to my memory, AFKT do consider mirror descent and study in any \ell_p space, I think it would be interesting to see the comparison of their work with this work.

**Summary Of The Paper:**

The paper is in the continuation of recent line of work that studies private algorithms when it has access to some public data. They also achieve a dimension independent bound as in some of the previous work. The idea of the paper is very simple: they use public data as the mirror map in the private mirror descent algorithm.

**Summary Of The Review:**

The approach in the paper is very natural and I think the algorithm would be very easy to deploy in large-scale system. The proof is very simple and elegant, which I think is another big bonus.

---

> ### Public Comment · ~Gautam_Kamath1 · 2021-11-09
> **Wrong score?**
>
> I'm wondering if the reviewer intended to give this paper a different score. The text seems very positive, calling the paper an easy accept, but the final score is currently a strong reject.

---

> > ### Comment · Reviewer_GaSe · 2021-11-09
> > **Score updated**
> >
> > Oh no! I made a mistake in entering the wrong score. It is a clear accept paper for me, so I will update the score.

---

> ### Author Response · Authors · 2021-11-14
> **Response to reviewer GaSe**
>
> Thank you for the supportive review. In the following we address the specific questions.
>
> 1. Q: On comparison (even theoretically) with prior work w.r.t. the sample complexity of public data: Our result states that for any convex ERM problem (and with an $\ell_2$-bounded constraint set $\mathcal{C}$), if one has p (dimensionality) public examples, then the excess risk bounds are independent of explicit dependence on p. To our knowledge, the only other paper that gives such a bound (without any additional conditions on the loss function (i.e., being in some low-rank subspace etc.)) is the result by Feldman et al. (https://arxiv.org/abs/1808.06651). As we mentioned in the paper, that particular result has a couple of drawbacks: i) the privacy analysis relies on convexity, and hence the algorithm is unusable in the empirical settings we consider in the paper (e.g., training neural networks), and ii) unlike Feldman et al., the number of public examples we need to get “dimension independence” can be much lower than p, if the loss function $\Psi$ is strongly convex with respect to the $Q$-norm for a set $Q$ that has small Gaussian width. In that sense, we improve on the SoTA.
> 2. Q: Empirical comparison to Asi et al.: We did not actually re-run their experiment (to avoid the issues of hyperparameter tuning). Instead, we ran experiments for our technique in a setting that was our best-effort to match the setting mentioned in Asi et al. We do realize that there can still be differences in the setups. We did reach out to the authors. The response was that they were still updating the code for public release. The github link (https://github.com/apple/ml-private-adaptive-gradient-methods) for the code in the published version of the paper is broken.
> 3. Comparing to other DP algorithms, that operate over other geometries, than euclidean geometry: In all the papers mentioned, we believe the geometric assumption appears as follows: if we bound the model space in the $\ell_p$-norm, then we bound the gradient space in the $\ell_q$-norm (where 1/p+1/q=1). Mirror maps which are strongly convex w.r.t. $\ell_p$-norm (and with appropriate noise addition) provide the best known population risks in those settings. In our current work though, we only assume that the gradients are bounded in $\ell_2$-norm. It is indeed an interesting question to understand the implications of our work w.r.t. the number of samples needed for dimension independence in settings where the gradients are bounded in other norms. We will add a more formal variant of this discussion in the paper.

---

### Decision · Program_Chairs · 2022-01-20

**Decision:**

Reject

**Comment:**

This paper proposes a new algorithm for private ERM, when given access to public data, with a dimension-independent risk guarantee if  (A) the public and private datasets are of the same distribution, (B) public dataset size exceeds the dimensionality (or, rather, the squared Gaussian width of an appropriate set), and (C) the public and private loss functions share a minimizer (and the gradients at the shared minimizer must satisfy some variance bounds). The algorithm uses the public data as the Bregman mirror map within private mirror descent (where Gaussian noise is added to the gradients), thus implicitly affecting the geometry, as opposed to explicitly learning the geometry as done in earlier works.

One reviewer was very positive, but two hovered around the borderline and expressed some reservations about the theory and experiments. Regarding the experiments, they did not compare to the ICML'21 paper by Asi et al --- however the authors of that paper have (surprisingly) still not released their code, so I think this is forgivable. Since the paper was on the borderline, I read it myself, with a focus on the theoretical aspects. I find myself agreeing with the second reviewer that the assumptions are strong, and their justification is weak and unrealistic.

Regardless of whether the paper, is accepted or not, I strongly recommend the authors to add condition (C) to their abstract (just the part about the shared minimizer) --- currently the abstract mentions two of the above but not the critical third one. I think (A) is already a strong assumption --- their justification that some users opt-in to reveal their data does not justify this, because the opt-in will not be random (if the opt-in depends on covariates like gender/age/..., the datasets will not be identically distributed). On top of that, (C) is also a strong assumption --- indeed usually the loss functions would be different (for eg, the private one would be clipped, and clipping will rarely preserve the population minimizer, as well as regularized) --- their justification that for a linear model with symmetric noise, clipping does not change the minimizer may be true (though not proved), but we would never expect the linear model to be true in practice even if we employ it as a working model. Last, assumption (B) restricts its use in many common high-dimensional data problems. Overall, I am pressed into a corner to find situations in which all three assumptions would be true.

Nevertheless, supposing that these assumptions hold, the algorithm is indeed clean, and the empirics appear reasonable. Overall, the paper remains on the borderline. Whether accepted or rejected, I expect the authors to do a much better job of carefully justifying their assumptions, with realistic and not far-fetched examples (as suggested by the second reviewer).